# Cortical dynein pulling mechanism is regulated by differentially targeted attachment molecule Num1

**Safia Omer[1], Samuel R Greenberg[2], Wei-Lih Lee[2]***

[1]Molecular and Cellular Biology Graduate Program, University of Massachusetts, Amherst, United States; [2]Department of Biological Sciences, Dartmouth College, Hanover, United States

**Abstract** Cortical dynein generates pulling forces via microtubule (MT) end capture-shrinkage and lateral MT sliding mechanisms. In *Saccharomyces cerevisiae*, the dynein attachment molecule Num1 interacts with endoplasmic reticulum (ER) and mitochondria to facilitate spindle positioning across the mother-bud neck, but direct evidence for how these cortical contacts regulate dynein-dependent pulling forces is lacking. We show that loss of Scs2/Scs22, ER tethering proteins, resulted in defective Num1 distribution and loss of dynein-dependent MT sliding, the hallmark of dynein function. Cells lacking Scs2/Scs22 performed spindle positioning via MT end capture-shrinkage mechanism, requiring dynein anchorage to an ER- and mitochondria-independent population of Num1, dynein motor activity, and CAP-Gly domain of dynactin Nip100/p150[Glued] subunit. Additionally, a CAAX-targeted Num1 rescued loss of lateral patches and MT sliding in the absence of Scs2/Scs22. These results reveal distinct populations of Num1 and underline the importance of their spatial distribution as a critical factor for regulating dynein pulling force.
DOI: https://doi.org/10.7554/eLife.36745.001

**\*For correspondence:**
wei.lih.lee@dartmouth.edu

**Competing interests:** The authors declare that no competing interests exist.

## Introduction

Proper positioning of the mitotic spindle is essential for successful cell division and is crucial for a wide range of processes including creation of cellular diversity during development, maintenance of adult tissue homeostasis, and balancing self-renewal and differentiation in progenitor stem cells (*Galli and van den Heuvel, 2008*; *Gómez-López et al., 2014*; *Morin and Bellaïche, 2011*; *Siller and Doe, 2009*). In various cell types, spindle positioning involves attachment of the minus end-directed MT motor cytoplasmic dynein to the cell cortex, where it exerts pulling force on astral MTs that emanate from the spindle poles (*di Pietro et al., 2016*; *Kotak and Gönczy, 2013*; *McNally, 2013*). While proteins involved in anchoring dynein have been identified (*Ananthanarayanan, 2016*; *Couwenbergs et al., 2007*; *Du and Macara, 2004*; *Heil-Chapdelaine et al., 2000*; *Kotak et al., 2012*; *Nguyen-Ngoc et al., 2007*; *Saito et al., 2006*; *Thankachan et al., 2017*) and the mechanism whereby dynein steps along the MT is becoming elucidated (*DeSantis et al., 2017*; *DeWitt et al., 2015*; *Grotjahn et al., 2018*; *Nicholas et al., 2015*; *Urnavicius et al., 2018*), how pulling forces are precisely regulated to achieve the appropriate spindle displacement remains incompletely understood.

The budding yeast *Saccharomyces cerevisiae* provides an important model for studying spindle position regulation [for review see (*Xiang, 2017*)]. During metaphase, the yeast spindle moves into the mother bud neck via dynein-dependent sliding of astral MT along the bud cortex (*Adames and Cooper, 2000*; *Moore et al., 2009*; *Yeh et al., 2000*). In the current model, dynein is recruited from the dynamic plus ends of astral MTs to cortical foci containing the attachment molecule Num1; once anchored, dynein uses its minus end-directed motor activity to walk along the MT lattice, generating

**eLife digest** Cells must divide so that organisms can grow, repair damaged tissues or reproduce. Before dividing, a cell creates two identical copies of its genetic information – one for each daughter. A molecular machine known as the mitotic spindle then moves each set of genetic material to where it will be needed when the daughter cells form. For the process to work properly, however, a motor protein known as dynein must correctly position the spindle by pulling it into place from the outskirts of the cell.

When a baker's yeast cell divides, it first forms a 'bump', which grows into a bud that will ultimately become another yeast. The spindle needs to be precisely placed at the midpoint between the original cell and the bud, so the genetic material can get into the future daughter cell. To do so, dynein travels to the bud, where a protein called Num1 helps it attach to the periphery and pull the filaments of the mitotic spindle (known as microtubules) to the correct position. Num1 also attaches to other cellular structures in the bud, including one known as the endoplasmic reticulum. It was unclear how this connection changes where dynein is located, and how it can pull on the spindle.

To study this, Omer et al. labeled Num1, dynein and microtubules with fluorescent markers so they could be followed in living baker's yeast using time-lapse microscopy. Mutant yeast strains were also used to disrupt how these proteins associate, which helps to tease out their roles. The experiments show that there are several populations of Num1 in the bud. One associates with the endoplasmic reticulum, and it helps dynein grab the side of a microtubule and make it slide into the bud. The other does not attach to the reticulum, but instead is located at the very tip of the bud. There, it makes dynein capture the end of the microtubule; this destabilizes the filament, which starts to shorten. As the microtubule shrinks, the spindle is pulled closer to the bud's tip, which aligns it in the right position. The yeast cells thus need Num1 in both locations to fine-tune the pulling activity of dynein, and the spindle's final positioning.

In the human body, not all divisions create two identical cells; for example, the daughters of stem cells can have different fates. This is due to a precise asymmetric division which dynein partly controls. The results by Omer et al. could help to unravel this mechanism.
DOI: https://doi.org/10.7554/eLife.36745.002

pulling forces on astral MTs along the bud cortex, thereby moving the connected spindle into the bud neck (*Lee et al., 2005, 2003*; *Markus et al., 2011*; *Sheeman et al., 2003*).

In contrast to the yeast model, studies in *C. elegans* embryos and mammalian cells show that cortically anchored dynein is able to mediate spindle movement by pulling on astral MTs in an apparent 'end-on' fashion (*Guild et al., 2017*; *Gusnowski and Srayko, 2011*; *Kiyomitsu and Cheeseman, 2012*; *Nguyen-Ngoc et al., 2007*; *Redemann et al., 2010*; *Schmidt et al., 2017*). Indeed, in vitro reconstitution studies using either bead-bound brain dynein or barrier-attached yeast dynein show that dynein can capture dynamic MT plus ends and generate pulling force on the captured MT (*Hendricks et al., 2012*; *Laan et al., 2012*). These experiments suggest that the particular geometry of the interaction between the barrier-attached dynein and the captured MT might promote MT shrinkage due to the barrier effect. Why 'capture-shrinkage' mechanism is not observed for Num1-based 'cortical pulling' has remained enigmatic. On the one hand, a classic study hinted that dynein pulls on the MT tips by inducing MT catastrophe at the cell cortex (*Carminati and Stearns, 1997*); on the other hand, a recent work suggested that dynein destabilizes astral MT plus ends regardless of their cortex interaction and that this activity might not be used for generating force for spindle movement (*Estrem et al., 2017*). Additionally, the MT-cortex interactions described by *Carminati and Stearns. (1997)* occurred before or after the nuclei moved into the neck, thus it is unknown whether they were mediated by the Num1-based mechanism that moves the spindle *into* the neck. Intriguingly, another study implicated cortical dynein in helping Bud6 (a cortical MT capture protein) and Bim1/EB1 (a plus end tracking protein) to couple shrinking MT plus ends to the cortex during an 'early' MT capture-shrinkage pathway mediated by the kinesin Kip3 (a MT plus end depolymerase) (*Ten Hoopen et al., 2012*). This study, however, shows that Num1 is not required for the 'early' MT capture-shrinkage pathway, which functions to mediate movement of the spindle pole

body (SPB) toward the incipient bud site. Together these data raise the question of whether dynein-mediated MT capture-shrinkage is downregulated during spindle movement into the bud neck.

Recent work suggests that organelles may also have an important role in regulating dynein function in spindle positioning. For example, mitochondria appear to drive the assembly of a subset of cortical Num1 patches, which in turn serve to anchor the organelle itself as well as dynein to the cell cortex (*Kraft and Lackner, 2017*). Num1 also appears to associate with cortical ER through interaction with the conserved ER membrane VAP (vesicle-associated membrane protein-associated protein), Scs2 (*Chao et al., 2014*; *Lackner et al., 2013*). In yeast, the VAP homologues Scs2 and Scs22 (hereafter abbreviated as Scs2/22) have been implicated in the formation of ER-PM tethering sites at the cell cortex (*Loewen et al., 2007*; *Manford et al., 2012*) and the ER diffusion barrier at the bud neck (*Chao et al., 2014*). The latter is important for limiting Num1 to the mother cell until M phase, thereby regulating the timing of dynein attachment in the bud compartment. However, the distribution and appearance of Num1 patches associated with ER, mitochondria, and PM appear to be different (*Chao et al., 2014*; *Heil-Chapdelaine et al., 2000*; *Klecker et al., 2013*; *Kraft and Lackner, 2017*; *Ping et al., 2016*; *Tang et al., 2009*), suggesting that dynein might be differentially regulated by different pools of Num1. Additionally, despite the identification of the organelles involved in Num1 recruitment, the nature of the MT-cortex interactions and the associated nuclear movements affected by each organelle remain unclear.

In this study, we set out to determine how changes in cortical Num1 localization alter dynein function, localization, and pulling mechanism in cells lacking the ER tether proteins Scs2/22. Consistent with previous work (*Chao et al., 2014*), we show that Num1 is concentrated in foci at polarized sites in *scs2/22Δ* cells, instead of being distributed throughout the cell cortex. We then show that the population of Num1 at the bud tip appears to be independent of mitochondria and is strikingly sufficient for dynein function in nuclear migration. We report direct observation of Num1- and dynein-dependent MT capture-shrinkage activity at the bud tip, explaining why nuclear migration across the bud neck can proceed as normal (albeit with a decreased efficiency) in the absence of classical dynein-mediated MT sliding along the bud cortex. The observed MT capture-shrinkage events require dynein anchoring at the bud tip and dynein motor activity, as well as MT tethering activity by the CAP-Gly domain of the Nip100/p150$^{Glued}$ subunit of dynactin, but not the MT plus end depolymerase activity of kinesin Kip3 or Kar3. Remarkably, defects in MT sliding in *scs2/22Δ* are corrected by a CAAX-targeted Num1, which restores lateral Num1 patches along the bud cortex and rescues the frequency of nuclear migration to WT level, highlighting a role for the ER-dependent population of cortical Num1. Our results suggest that, in situations where cortical pulling forces drive cellular positioning processes, spatial distribution of dynein attachment molecule could potentially offer a mechanism to regulate dynein pulling force by influencing the relative activity of lateral versus end-on dynein contacts with MT at the cell cortex.

## Results

### Loss of Scs2/22 disrupts Num1 localization and reveals a distinct pool of Num1 at the polarized cell ends

In WT cells, Num1 forms dim and bright patches throughout the cell cortex (*Figure 1A*; *Video 1*, top) (*Heil-Chapdelaine et al., 2000*; *Tang et al., 2009*). We found that cells lacking both cortical ER tethers Scs2 and Scs22 exhibited a dramatic loss of dim Num1 patches (*Figure 1A*; *Video 1*, bottom) and a significant reduction in the number of bright Num1 patches (*Figure 1B*). More than 70.0% of *scs2/22Δ* budded cells displayed ≤2 bright patches compared to only 6.0% in WT budded cells. The remaining Num1 patches in *scs2/22Δ* were observed as stationary foci at the polarized ends of the cell (i.e. the distal bud tip and the mother cell apex; *Figure 1A and C*) and as motile foci in the cytoplasm (*Figure 1—figure supplement 1A*). Loss of Scs2 alone had a similar effect, whereas loss of Scs22 alone had no effect (*Figure 1—figure supplement 1B–D*). However, loss of both proteins was worse than the loss of Scs2 alone (*Figure 1—figure supplement 1C*; 2.03 ± 1.1 versus 2.8 ± 1.3 patches per cell for *scs2/22Δ* and *scs2Δ*, respectively), suggesting that Scs22 may have a redundant role when Scs2 is absent. Thus, we carried out all subsequent analysis in the *scs2/22Δ* double mutant background.

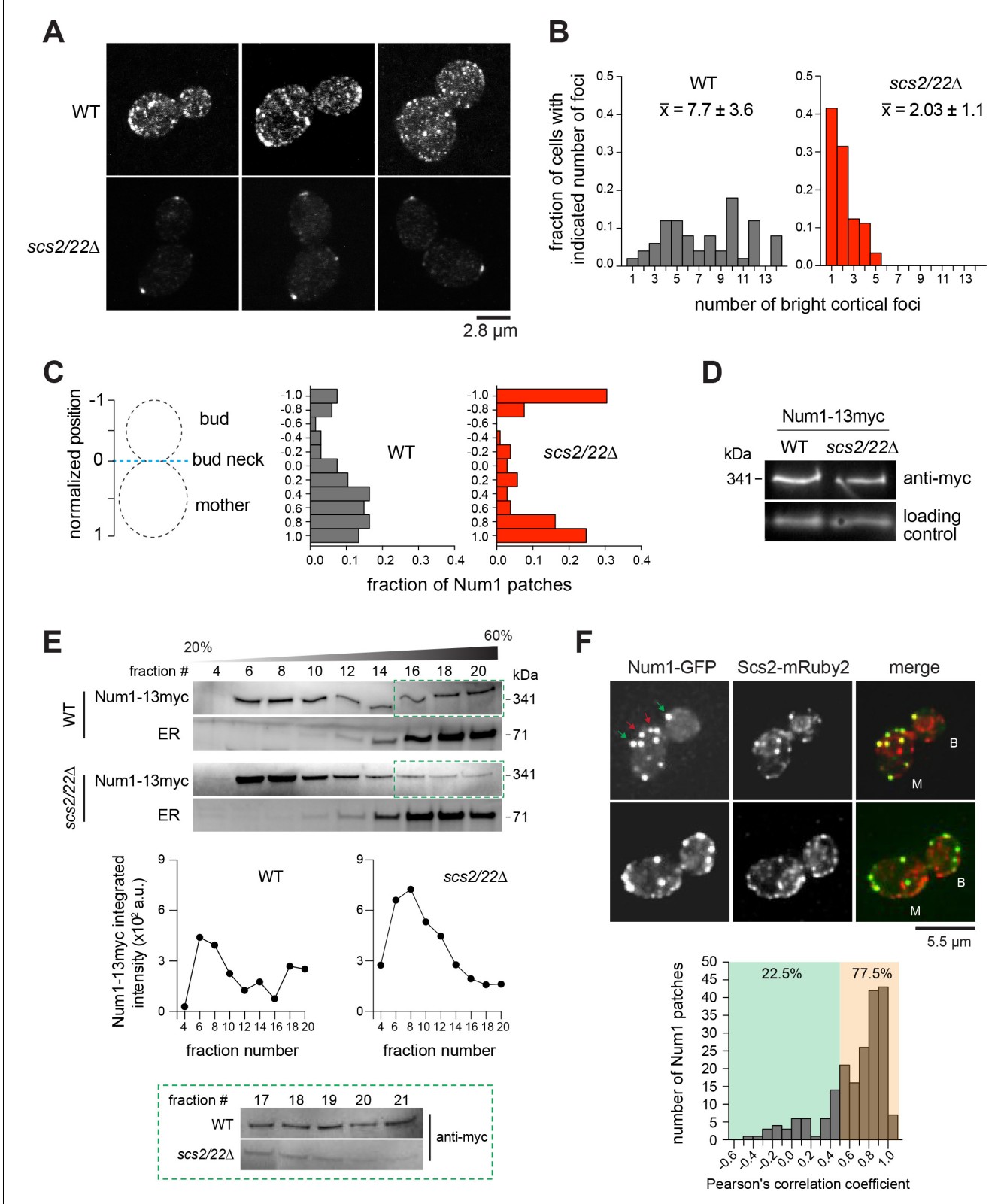

**Figure 1.** Num1 localization is altered by deletion of Scs2/22. (**A**) 2D projections of 3D confocal stack images of Num1-GFP in WT and *scs2/22Δ* cells. (**B**) Fraction of cells with indicated number of Num1-GFP patches. x̄, average number of patches per cell (n ≥ 50 cells per strain). (**C**) Distribution of Num1-GFP patches along the cortex. The position of each patch was projected on the mother-bud axis and normalized to the bud neck. Positive distances indicate that the patch was in the mother cell, whereas negative distances indicate that the patch was in the daughter cell (n = 46 and 16 cells

*Figure 1 continued on next page*

*Figure 1 continued*

for *scs2/22Δ* and WT, respectively). (**D**) Western blots showing Num1-13myc levels in whole cell lysates of indicated strains. (**E**) Sucrose gradient sedimentation analysis of Num1-13myc in WT and *scs2/22Δ* strains. Whole cell lysates from each strain were loaded onto 20-60% sucrose gradients, sedimented, and analyzed by Western blot using anti-c-Myc (for Num1-13myc) and anti-Sac1 (for ER) antibodies. *Top*, representative sedimentation profiles from two independent experiments. *Middle*, Num1-13myc band intensity plotted against fraction number. *Bottom*, Western blot showing Num1-13myc in fractions 17 through 21. (**F**) Deconvolved wide-field images of Num1-GFP and Scs2-mRuby2 in WT cells. Each image is a 2D projection of 11 optical sections spaced 0.5 μm apart. Green and red arrows indicate Num1-GFP patches that do and do not colocalize with Scs2-mRuby2 foci, respectively. B, bud; M, mother. *Bottom*, histogram of Pearson's correlation coefficients for the colocalization of Num1-GFP with Scs2-mRuby2 (n = 200 cortical Num1 patches found in either bud or mother cell).

DOI: https://doi.org/10.7554/eLife.36745.003

The following figure supplements are available for figure 1:

**Figure supplement 1.** Num1-GFP localization in *scs2Δ* and *scs22Δ* single mutants and *scs2/22Δ* double mutant.
DOI: https://doi.org/10.7554/eLife.36745.004
**Figure supplement 2.** FRAP of Num1-GFP foci in WT and *scs2/22Δ* cells.
DOI: https://doi.org/10.7554/eLife.36745.005
**Figure supplement 3.** Deletion of Scs2/22 but not Num1 results in loss of cortical ER.
DOI: https://doi.org/10.7554/eLife.36745.006
**Figure supplement 4.** Time-lapse images of WT and *scs2/22Δ* cells expressing Num1-GFP.
DOI: https://doi.org/10.7554/eLife.36745.007
**Figure supplement 5.** Num1-GFP clustering in *scs2/22Δ* is independent of mitochondria segregation into buds.
DOI: https://doi.org/10.7554/eLife.36745.008

We asked whether Num1 stability is affected in *scs2/22Δ* cells. Immunoblot analysis revealed that Num1-13myc levels in *scs2/22Δ* were similar to WT cells (*Figure 1D*). Additionally, whole-cell intensity measurements showed that Num1-GFP levels were quantitatively the same as WT (*Figure 1—figure supplement 1E*). However, the mean intensity of individual Num1-GFP patches was approximately 2–3 folds higher in *scs2/22Δ* compared to WT (*Figure 1—figure supplement 1F*). Thus, loss of Scs2/22 affected Num1 distribution along the cell cortex but not Num1 stability.

We next examined whether Num1 mobility is affected in *scs2/22Δ* cells. FRAP analysis showed that cortical Num1-GFP patches in *scs2/22Δ* exhibited no fluorescence recovery after photobleaching (*Figure 1—figure supplement 2*), indicating that Num1-GFP was stably associated with the cortex, a result similar to that in WT cells (*Chao et al., 2014*; *Kraft and Lackner, 2017*). Additionally, although deletion of Scs2/22 resulted in a severe loss of cortical ER (*Figure 1—figure supplement 3A*) (*Loewen et al., 2007*; *Manford et al., 2012*), the timing for the accumulation of Num1 at the bud tip appeared to be unaffected compared to WT cells, as evident by imaging of single cells over time during bud growth (*Figure 1—figure supplement 4*). Conversely, no significant loss in cortical ER was observed in *num1Δ* cells (*Figure 1—figure supplement 3B*). Importantly, no effect on Num1-GFP clustering at the bud tip was observed in *scs2/22Δ* when mitochondrial segregation into the bud was disrupted by the single *mmr1Δ* or double *mmr1Δ gem1Δ* mutation (*Figure 1—figure supplement 5*) (*Frederick et al., 2008*), which contradicts the model in which Num1 clustering in the bud depends on mitochondrial inheritance (*Kraft and Lackner, 2017*). Our data suggest that localization of Num1 to polarized bud tips does not require Scs2/22 and mitochondria.

To assess whether the Num1 population distributed along the cell cortex was associated with ER, we analyzed sedimentation profiles of Num1-13myc in sucrose density gradients and colocalization of Num1-GFP with Scs2-mRuby2. Sucrose

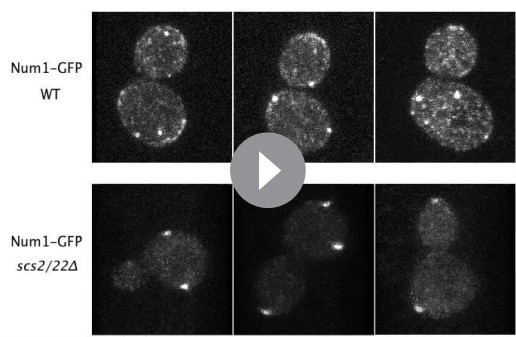

**Video 1.** Loss of Scs2/22 alters Num1 distribution along the cell cortex. Full 3D reconstructions of confocal stacks showing Num1-GFP localization in single WT (top row) and *scs2/22Δ* (bottom row) cells. Each stack consists of 18 optical sections spaced 0.3 μm apart encompassing the entire thickness of the cell.
DOI: https://doi.org/10.7554/eLife.36745.009

gradient sedimentation analysis showed that a pool of Num1-13myc co-fractionated with ER in an Scs2/22-dependent manner (*Figure 1E*). Colocalization analysis revealed that most Num1-GFP patches (155 out of 200; 77.5%) exhibited intensities that were correlated with the signal intensities of Scs2-mRuby2 (*Figure 1F*; $0.5 \leq$ Pearson's correlation coefficient $\leq 1$). However, a minority of Num1-GFP patches (45 out of 200; 22.5%) did not co-localize with Scs2-mRuby2 (Pearson's correlation coefficient < 0.5). These results, when combined with our analysis of Num1 localization in *scs2/22Δ* cells, implicate the existence of distinct populations of Num1 patches at the cell cortex.

## A small number of Num1 patches is sufficient for dynein pathway function

Next, we asked whether the observed change in Num1 localization in *scs2/22Δ* affects dynein targeting and function, as would be expected if Num1 functions as a cortical anchor for dynein. In WT cells, Dyn1-3GFP localizes to the SPB, astral MT plus ends, and to cortical foci where it has been off-loaded from the MT plus ends (*Lee et al., 2003*; *Sheeman et al., 2003*). In *scs2/22Δ* cells, we observed that Dyn1-3GFP localized similarly to the SPB and astral MT plus ends (*Figure 2A*) but the levels of Dyn1-3GFP at the MT plus ends were significantly enhanced compared to WT cells (*Figure 2B*), consistent with a reduced number of available offloading sites. In accord with the change in Num1 localization, cortical Dyn1-3GFP foci were found at the bud tip and mother apex of *scs2/22Δ* cells (*Figure 2—figure supplement 1A*). However, the mean fluorescence intensity of individual cortical Dyn1-3GFP foci was enhanced in *scs2/22Δ* relative to WT (2.1 and 3.1-fold higher for cortical foci found in the bud and mother, respectively; *Figure 2C*). A similar enhancement was observed for Jnm1-3mCherry (dynactin p50$^{dynamitin}$ subunit) at the MT plus ends and cortex (*Figure 2D* and *Figure 2—figure supplement 1B*). The difference in dynein targeting between *scs2/22Δ* and WT could not be attributed to changes in the expression level or the stability of dynein or dynactin (which is required for dynein-offloading), as determined by immunoblotting (*Figure 2E*). Furthermore, in *scs2/22Δ* cells, as reported for WT cells (*Markus et al., 2011*; *Moore et al., 2008*), plus end targeting of Dyn1-3GFP depended on Pac1/LIS1 (*Figure 2—figure supplement 1C*), and cortical targeting of Dyn1-3GFP depended on dynactin (*Figure 2—figure supplement 1C*), suggesting that regulation of dynein targeting remains intact even though dynein anchoring is limited to the polar ends of the cell.

We first assessed dynein pathway function using a single-time point spindle orientation assay. Strikingly, *scs2/22Δ* strain had only 0.7% of cells with a misoriented anaphase spindle phenotype, quantitatively similar to that observed for WT (0.9%; *Figure 2F*), indicating that dynein pathway is functional. In contrast, *scs2/22Δ* strain expressing Num1$^{L167E+L170E}$ (hereafter referred to as Num1$^{LL}$), which harbors two point mutations that abolish the Num1-dynein interaction but does not interfere with the Num1 cluster formation (*Figure 2—figure supplement 1D and E*) (*Tang et al., 2012*), exhibited a high level of misoriented anaphase spindle phenotype (42.6%; *Figure 2F*) similar to that observed for a *dyn1Δ* or *num1Δ* strain (40.2 and 48.2%, respectively; *Figure 2F*), indicating that Num1-dynein interaction is required for proper spindle orientation in the *scs2/22Δ* background. The same results were obtained when nuclear segregation was assayed by DAPI staining (*Figure 2—figure supplement 1F*). These data demonstrate that the remaining Num1 patches in *scs2/22Δ*, albeit few in number, appear to be sufficient for dynein pathway function.

We further assessed dynein function by assaying for synthetic growth defects with *kar9Δ* and *cin8Δ*. Budding yeast lacking Kar9 or Cin8 requires the dynein pathway for normal growth (*Geiser et al., 1997*; *Gerson-Gurwitz et al., 2009*; *Miller and Rose, 1998*). Tetrad dissection analysis revealed that *scs2/22Δ kar9Δ* and *scs2/22Δ cin8Δ* triple mutant progeny formed viable colonies, exhibiting no growth defects when compared with *scs2/22Δ* double mutant (*Table 1*), consistent with the dynein pathway being functional in *scs2/22Δ*. Additionally, no synthetic effect on growth was observed for triple mutant of *scs2/22Δ* with *dyn1Δ* (*Table 1*). These genetic data further support the notion that the residual Num1 patches in *scs2/22Δ* cells are sufficient for dynein pathway function.

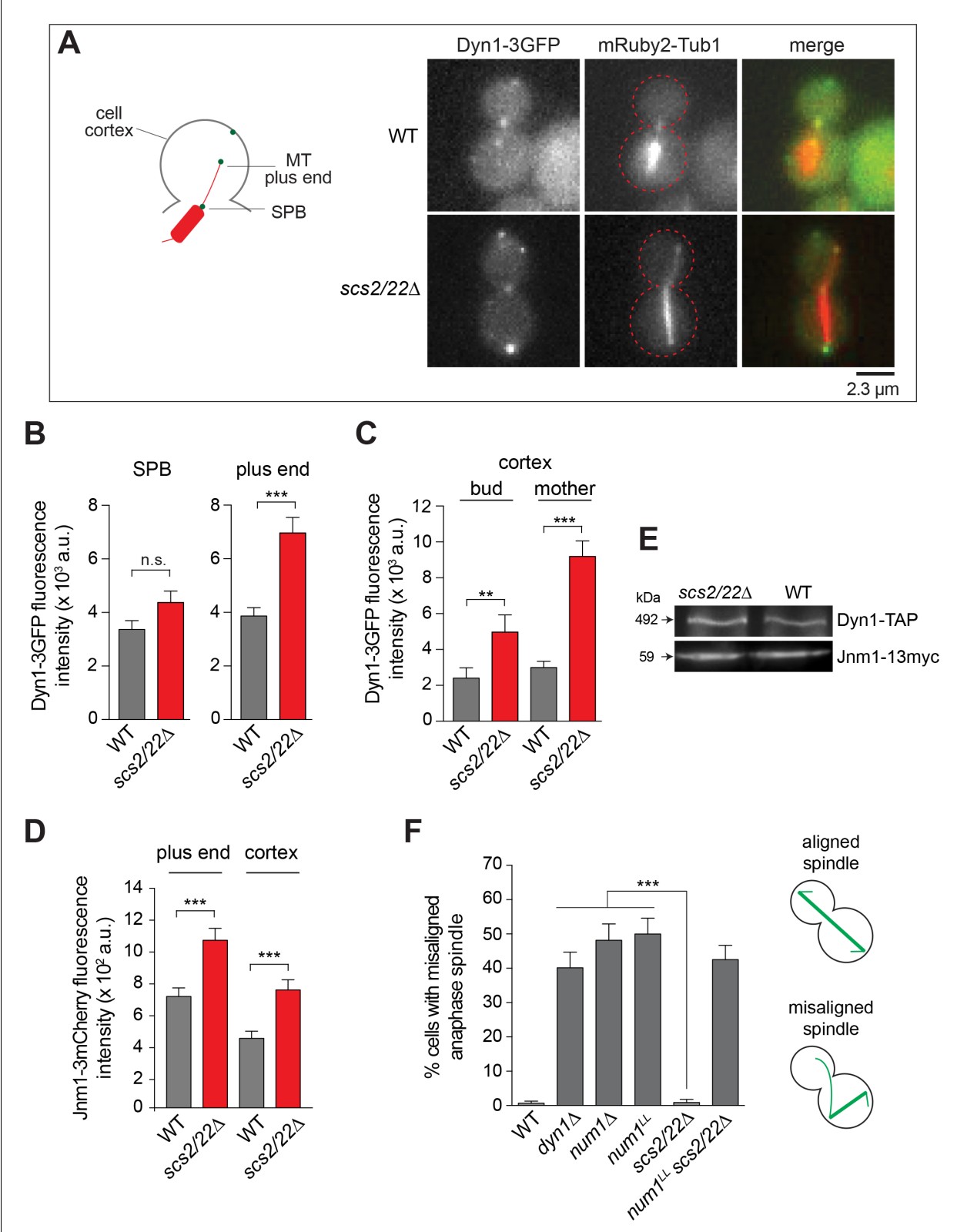

**Figure 2.** Dynein localization and function in *scs2/22Δ* cells. (A) Wide-field images of live cells expressing Dyn1-3GFP and mRuby2-Tub1 in WT and *scs2/22Δ* cells. (B and C) Dyn1-3GFP fluorescence intensity at the SPB (n ≥ 32), plus end (n ≥ 60), and cortex (n ≥ 110). Error bars depict the standard error of the mean (SEM). n.s., not statistically significant; **p<0.005; ***p<0.0001 by unpaired *t* test. (D) Jnm1-3mCherry fluorescence intensity at the plus end (n ≥ 50) and cortex (n ≥ 29). Error bars indicate SEM. ***p<0.0001 by unpaired *t* test. (E) Western blots of Dyn1-TAP and Jnm1-13myc levels in

*Figure 2 continued on next page*

*Figure 2 continued*

total cell lysates of indicated strains. (F) Percentage of misaligned anaphase spindle (n > 110 for each strain). Error bars indicate the standard error of proportion (SEP). ***p<0.0001 by one-way ANOVA test.

DOI: https://doi.org/10.7554/eLife.36745.010

The following figure supplement is available for figure 2:

**Figure supplement 1.** Dynein and dynactin localization in *scs2/22Δ*.

DOI: https://doi.org/10.7554/eLife.36745.011

## Dynein mediates spindle movements via capture-shrinkage of astral MTs at the bud tip

Given the dramatic change in Num1 localization, we wondered how dynein would mediate spindle positioning in *scs2/22Δ* cells. We assessed dynein-dependent spindle movements by assaying for anaphase spindle re-alignment from a misoriented position, hereafter referred to as spindle correction (*Yeh et al., 2000*). Kar9 was deleted to increase the frequency of spindle misalignment and to enhance dynein-dependent spindle movements (*Moore et al., 2009*; *Yeh et al., 2000*). The mechanism of spindle correction was scored based on time-lapse images of astral MT interaction with the bud cortex (as detailed in Materials and methods). In *kar9Δ* cells, spindle correction was predominantly mediated by MT sliding along the bud cortex (86.7%, n = 30 events; *Figure 3A and B*; *Video 2*, top), as previously reported (*Adames and Cooper, 2000*; *Yeh et al., 2000*). In contrast, in *scs2/22Δ kar9Δ* cells, we observed that spindle correction was primarily mediated by capture-shrinkage of the astral MT plus end at the bud tip (77.8%, n = 63 events; *Figure 3A and B*). Two-color movies of mRuby2-Tub1 and Num1-GFP revealed that capture-shrinkage of the astral MT occurred upon 'end-on' interaction of the plus end with a Num1 patch at the bud tip (*Video 2*, bottom). Notably, the plus end stayed in contact with the Num1 patch while shrinking, pulling the minus-end-attached spindle into the bud, causing spindle correction. In separate experiments, we acquired movies with a larger number of optical sections confirming that the astral MT did not slide over the surface of the bud tip (*Video 3*; displayed as XY and XZ frames). The same capture-shrinkage phenomenon was also observed when hydroxyurea (HU)-arrested preanaphase *scs2/22Δ kar9Δ* cells were examined. These data indicate that the change in the distribution of cortical Num1 in *scs2/22Δ* cells has apparently altered the mechanism of dynein-mediated spindle positioning.

Kymograph analysis of MT capture-shrinkage events revealed that Dyn1-3GFP persisted at the shrinking MT plus end contacting the bud tip (15 out of 16 events; *Figure 3C*; *Video 4*, top), supporting the idea that dynein is involved in generating the cortex-coupled pulling force during MT depolymerization at the Num1 site. Consistent with this notion, loss of Dyn1 abolished spindle correction in *scs2/22Δ* cells (0 out of 138 spindles were corrected; *Figure 3B* and *Figure 3—figure supplement 1A*).

We considered the possibility that other MT plus end depolymerases might also be involved in force generation at the Num1 site. However, we found that the frequency of observing MT capture-shrinkage was unaffected in *scs2/22Δ kar9Δ* cells lacking Kip3 (kinesin-8) or Kar3 (kinesin-14)

**Table 1.** Viability of *scs2/22Δ* mutant in combination with *kar9Δ*, *dyn1Δ*, or *cin8Δ* mutant.

| Mutant combination | Number of tetrads analyzed | Number of predicted double or triple mutants | Viability of mutants | |
|---|---|---|---|---|
| | | | Viable | Microcolony |
| *scs2/22Δ* | 22 | 16 | 16 | 0 |
| *scs2/22Δ kar9Δ* | 13 | 5 | 5 | 0 |
| *scs2/22Δ dyn1Δ* | 22 | 9 | 9 | 0 |
| *scs2/22Δ cin8Δ* | 29 | 17 | 17 | 0 |
| *dyn1Δ cin8Δ* | 10 | 7 | 0 | 7 |

For *scs2/22Δ* and *scs2/22Δ dyn1Δ* combinations, YWL4865 was crossed with YWL521. For *scs2/22Δ kar9Δ* combination, YWL4865 was crossed with YWL4949. For *scs2/22Δ cin8Δ* and *dyn1Δ cin8Δ* combinations, YWL3955 was crossed with YWL4866 and YWL504, respectively. The resulting diploid strains were sporulated and tetrads were dissected.

DOI: https://doi.org/10.7554/eLife.36745.012

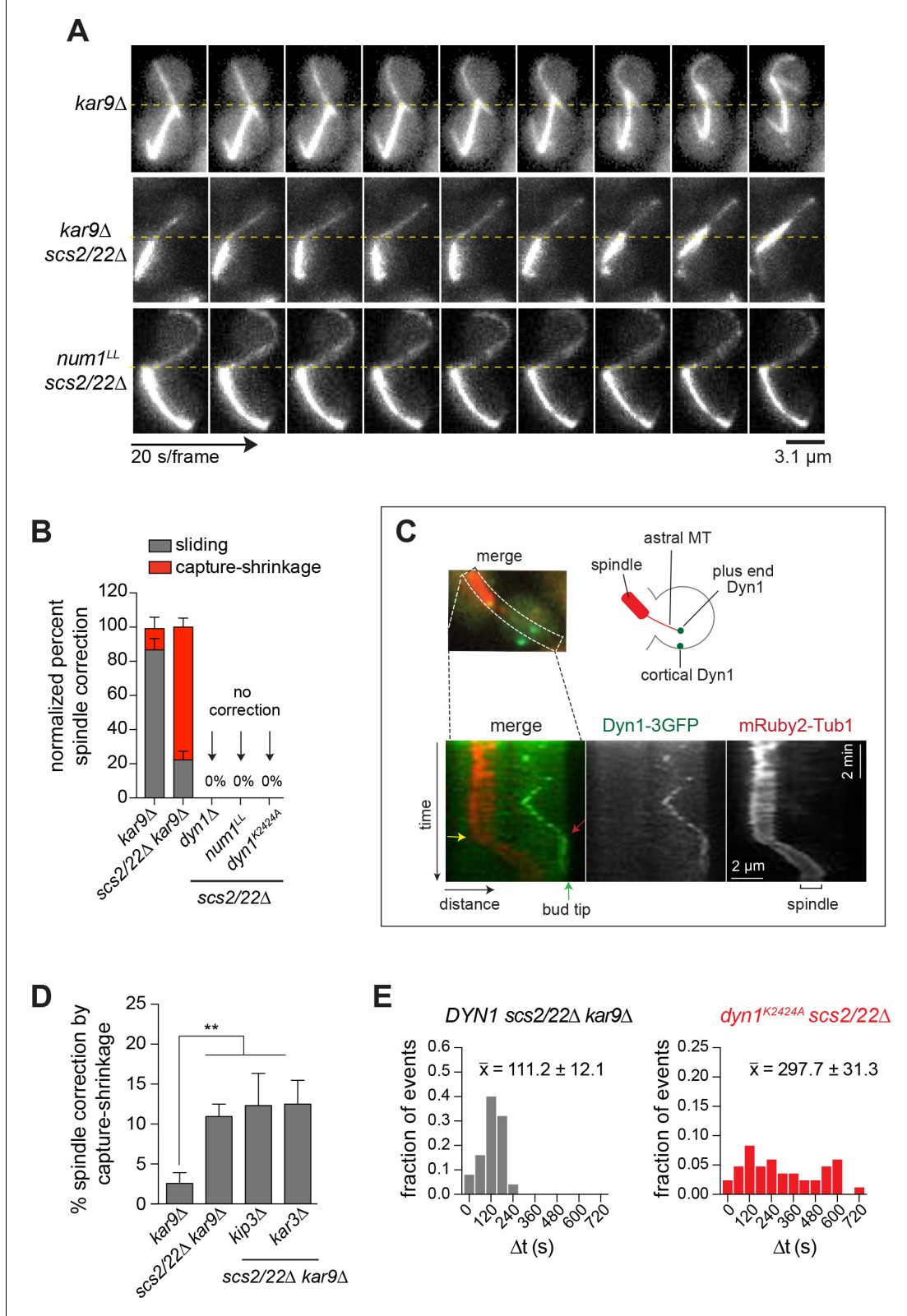

**Figure 3.** Dynein mediates spindle correction via capture-shrinkage mechanism in *scs2/22Δ* cells. (**A**) Representative movie frames of mRuby2-Tub1 showing dynein-dependent spindle correction via sliding or capture-shrinkage mechanism in *kar9Δ* and *scs2/22Δ kar9Δ* cells during a 10-min movie. No spindle correction was observed in *num1^LL scs2/22Δ*. Dashed line marks the bud neck position. (**B**) Quantification of spindle correction mechanisms for *kar9Δ* (n = 30), *scs2/22Δ kar9Δ* (n = 63), *dyn1Δ scs2/22Δ* (n = 138), *num1^LL scs2/22Δ* (n = 91), and *dyn1^K2424A scs2/22Δ* (n = 99). Error bars depict SEP. (**C**)
*Figure 3 continued on next page*

*Figure 3 continued*

Kymograph analysis of Dyn1-3GFP and mRuby2-Tub1 in *scs2/22Δ kar9Δ* showing persistence of dynein at the bud tip during a MT shrinkage event. *Green arrow*, position of bud tip; *red arrow*, initial contact of Dyn1-3GFP with the bud tip; *yellow arrow*, start of spindle movement. (D) Percentage of spindle correction by end-on capture-shrinkage mechanism in *kar9Δ*, *scs2/22Δ kar9Δ*, *kar3Δ scs2/22Δ kar9Δ*, and *kip3Δ scs2/22Δ kar9Δ* during a 10-min movie. Error bars indicate SEP (n ≥ 65). \*\*p<0.005 by unpaired *t* test. (E) Histogram of the duration of plus end attachment at the bud tip (Δt) for each indicated strain. $\bar{x}$, average duration of contact (n ≥ 25 per strain).

DOI: https://doi.org/10.7554/eLife.36745.013

The following figure supplement is available for figure 3:

**Figure supplement 1.** Dynein motor activity is required for capture-shrinkage of astral MT plus ends in *scs2/22Δ*.

DOI: https://doi.org/10.7554/eLife.36745.014

(*Figure 3D*; *Video 5*, top), two kinesin motors with known plus end depolymerase activity (*Gupta et al., 2006*; *Sproul et al., 2005*), indicating that these motors are not responsible for the capture-shrinkage phenomenon seen at the Num1 site. On the other hand, we found that disrupting dynein-anchoring using the *num1^{LL}* allele abolished MT capture-shrinkage and prevented spindle correction (0 out of 91 spindles were corrected; *Figure 3A and B*; *Video 4*, bottom). In *num1^{LL} scs2/22Δ* cells, no capture-shrinkage events occurred despite the fact that astral MT plus ends with accumulated Dyn1-3GFP were seen sweeping along the bud tip (*Video 4*, bottom), indicating that cortical anchoring is required for dynein to generate the cortex-coupled pulling force at the Num1 site. These results contradict a previous study postulating that dynein does not need to attach to the cortex to destabilize MT ends (*Estrem et al., 2017*).

Moreover, we noted that the mean astral MT length in *num1^{LL} scs2/22Δ* was not only longer than in *scs2/22Δ kar9Δ*, but also quantitatively the same as in *dyn1Δ scs2/22Δ* (*Figure 3—figure supplement 1B*), which further supports the model in which dynein acts as a MT destabilizer at the cortical Num1 site.

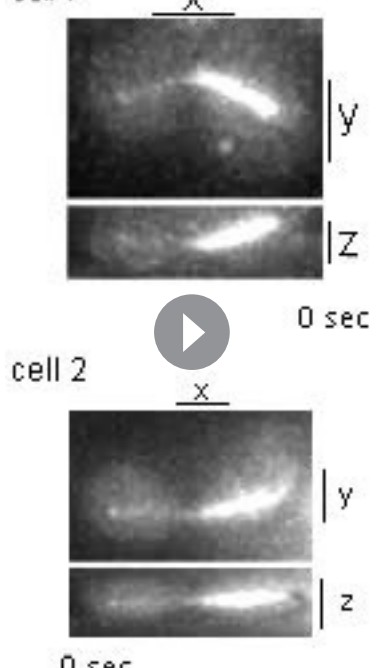

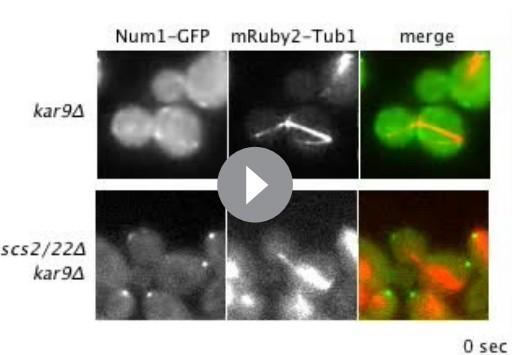

**Video 2.** Loss of Scs2/22 alters dynein pulling mechanism during spindle correction. Num1-GFP (*left*) and mRuby2-Tub1 (*middle*) showing lateral MT sliding (*kar9Δ*, top) and end-on MT capture-shrinkage at a Num1 site (*scs2/22Δ kar9Δ*, bottom) as the spindle translocates into the bud neck during spindle correction. Merge image shows MT in red and Num1 in green. Each frame is a maximum intensity projection of 3 optical sections spaced 0.5 μm apart. Movie was captured at 10 s intervals.

DOI: https://doi.org/10.7554/eLife.36745.015

**Video 3.** Astral MT undergoes capture-shrinkage but not sliding at the bud tip in *scs2/22Δ kar9Δ* cells. Full 3D time-lapse images displayed in XY and XZ views showing end-on interaction of the astral MT plus end in *scs2/22Δ kar9Δ* cells. Each frame is a maximum intensity projection of 7 μm (cell 1) or 9 μm (cell 2) optical sections spaced 0.5 μm apart. Movie was captured at 5 or 7 s intervals.

DOI: https://doi.org/10.7554/eLife.36745.016

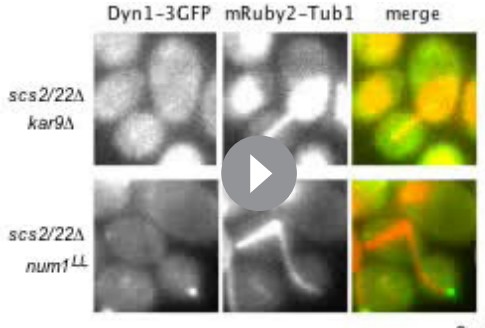

**Video 4.** Dynein anchorage at the bud tip is required for end-on capture-shrinkage of astral MT. *Top*, Dyn1-3GFP persists at the bud tip cortex during shrinkage of a captured astral MT plus end in a *scs2/22Δ kar9Δ* cell. *Bottom*, Dyn1-3GFP accumulates at the MT plus end but fails to attach to the bud tip to mediate MT capture-shrinkage in a *num1^LL^ scs2/22Δ* cell. Each frame is a maximum intensity projection of 3 optical sections spaced 0.5 μm apart. Movie was captured at 10 s intervals.

DOI: https://doi.org/10.7554/eLife.36745.017

To investigate whether dynein's motor activity is required for MT capture-shrinkage at the Num1 site, we made use of the established $dyn1^{K2424A}$ mutant, in which ATP binding was inhibited by a point mutation in the Walker A motif of the AAA3 domain (*Reck-Peterson and Vale, 2004*). We found that, while dynein targeting to the plus ends and cortex was not significantly affected (*Figure 3—figure supplement 1C*), spindle correction was abolished by the $dyn1^{K2424A}$ mutation in *scs2/22Δ* cells (0 out of 99 spindles were corrected; *Figure 3B* and *Figure 3—figure supplement 1A*), indicating that motor activity is required for the production of cortex-coupled pulling forces. Notably, in $dyn1^{K2424A}$ *scs2/22Δ* cells, astral MT plus ends that grew into the bud appeared to remain stably attached upon reaching the bud tip (*Video 5*, bottom). To quantitate MT attachment at the bud tip, we tracked the position of the plus ends over time (see Materials and methods). Compared with WT *DYN1*, $dyn1^{K2424A}$ mutant increased the duration of attachment by nearly three folds ($\Delta t = 111.2 \pm 12.1$ vs. $297.7 \pm 31.3$ s, n $\geq$ 25 for each; *Figure 3E*). Despite having a prolonged end-on interaction with the bud tip, the astral MTs were never observed to undergo shrinkage that led to a productive spindle movement. Conversely, we often observed the attached MTs to continue to grow and buckle while their plus ends stayed in contact with the bud tip (69.2% of $dyn1^{K2424A}$ *scs2/22Δ* cells compared to 3.8% of *DYN1 scs2/22Δ kar9Δ* cells exhibited buckling phenotype, n $\geq$ 52 cells for each). These observations suggest that dynein's motor activity is needed to destabilize MT plus ends at the bud tip, possibly by enhancing catastrophes, as previously suggested by in vitro studies (*Laan et al., 2012*). It is possible, however, that dynein's motor activity is only needed to maintain a dynamic connection between the MT plus end and the cortex at the bud tip, as the work to pull the spindle may be performed entirely by the shrinking MTs themselves (*Grishchuk et al., 2005*; *Kozlowski et al., 2007*).

## CAP-Gly domain of Nip100/p150^Glued^ is required for dynein-mediated capture-shrinkage of astral MTs

Next, we examined how dynactin might be required for the observed MT capture-shrinkage events at the bud tip Num1 site. The vertebrate p150^Glued^ subunit of dynactin contains a CAP-Gly domain and a basic region, both of which have been shown to bind MTs and enhance the processivity of dynein in vitro (*Ayloo et al., 2014*; *Culver-Hanlon et al., 2006*; *King and Schroer, 2000*; *Kobayashi et al., 2006*; *Waterman-Storer et al., 1995*). MT tethering by these domains might prevent dynein from dissociating from a shrinking MT end during capture-shrinkage events at the cortex (*Figure 4A*). To test this,

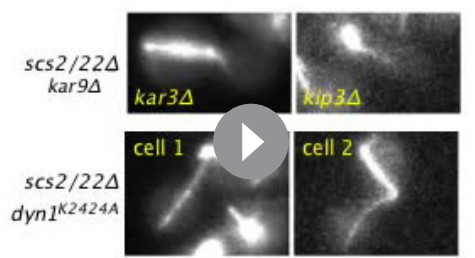

**Video 5.** Dynein motor activity but not Kip3 or Kar3 is required for MT capture-shrinkage. *Top*, movement and realignment of anaphase spindle into the bud neck upon end-on interaction of the astral MT plus end with the bud tip in a *kar3Δ scs2/22Δ kar9Δ* or *kip3Δ scs2/22Δ kar9Δ* cell. *Bottom*, captured astral MT plus ends fail to undergo shrinkage at the bud tip to mediate spindle realignment in *dyn1^K2424A^ scs2/22Δ* cells. Each frame is a maximum intensity projection of 3 optical sections spaced 0.5 μm apart. Movie was captured at 10 s intervals.

DOI: https://doi.org/10.7554/eLife.36745.018

we excised codons 2–103 from the endogenous *NIP100* gene, the budding yeast homologue of p150^Glued, to remove the analogous CAP-Gly and basic region of the protein (*Figure 4B*). To assess how capture-shrinkage was affected, we recorded time-lapse movies of spindle correction in *kar9Δ* background, as above. The number of *scs2/22Δ kar9Δ* cells, in which spindle correction occurred via MT capture-shrinkage at the bud tip, was dramatically decreased by the truncated Nip100 (*Figure 4C*). The reduction could not be attributed to a defect in MT growth toward the cell cortex, as we often observed astral MTs grow into the bud, making frequent contacts with the cell cortex (*Video 6*). Also, the average length of astral MTs was quantitatively the same for *scs2/22Δ kar9Δ* cells expressing the truncated or full-length version of Nip100 (*Figure 4D*), indicating that loss of the CAP-Gly domain did not affect the stability of astral MTs. Additionally, immunoblot analysis showed that the truncation did not affect the expression level of Nip100 (*Figure 4B*), indicating that the observed reduction in capture-shrinkage events could not be attributed to an overall reduction in protein stability.

To examine the contribution of the CAP-Gly domain more closely, we tracked the position of the astral MT plus ends in the time-lapse movies and quantitated their interaction with the bud tip. In *CAP-GlyΔ scs2/22Δ kar9Δ* cells, we observed the plus ends to interact with the bud tip for a significantly shorter duration compared with *scs2/22Δ kar9Δ* cells expressing the full-length Nip100 ($\Delta t = 55.2 \pm 7.2$ vs. $111.2 \pm 12.1$ s, $n \geq 21$ for each; *Figure 4E* and *Figure 4—figure supplement 1A*). In these abbreviated interactions, we could sometimes observe how an astral MT plus end, after making contact with the bud tip, underwent a brief capture-shrinkage event (coupled with SPB movement) that was suddenly aborted by its release from the cortex (*Video 6*, cell 2). This suggests that the MT tethering activity of the CAP-Gly domain of Nip100 is needed for the persistence of dynein-dependent MT capture-shrinkage events that power spindle correction through the bud neck.

As an alternative, we considered whether decreased spindle correction could be due to poor localization of dynein to the bud tip Num1 site, which we have shown above to be necessary for MT capture-shrinkage events. In *CAP-GlyΔ scs2/22Δ* cells, we observed that the localization of Dyn1-3GFP to the astral MT plus ends and the bud tip was unaffected (*Figure 4—figure supplement 1B*). We also quantitated the fluorescence intensity of individual foci and found that the amount of dynein per cortical focus was quantitatively the same as that observed in *scs2/22Δ* cells expressing the full-length Nip100 (*Figure 4—figure supplement 1C*). Thus, the decrease in MT capture-shrinkage caused by the *CAP-GlyΔ* mutation was not due to defective anchoring of dynein to the cortical contact point.

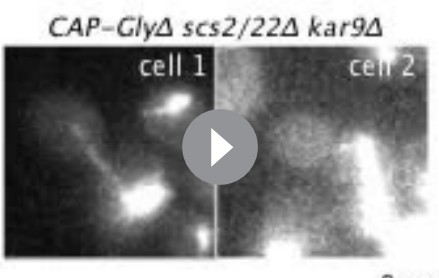

**Video 6.** CAP-Gly domain is required for MT tethering during capture-shrinkage of astral MT plus end. Movie frames of *CAP-GlyΔ scs2/22Δ kar9Δ* cells expressing mRuby2-Tub1 showing failure to initiate MT capture-shrinkage at the bud tip (cell 1) or failure to maintain stable interaction between the plus end and the bud tip during a MT capture-shrinkage event (cell 2). Each frame is a maximum intensity projection of 3 optical sections spaced 0.5 μm apart. Movie was captured at 10 s intervals.

DOI: https://doi.org/10.7554/eLife.36745.021

Our results thus far suggest that dynein has two modes of cortical pulling mechanisms for controlling spindle movement into the bud cell compartment. To examine the relationship between these two modes, we quantitated the extent and consequence of losing dynein function in *CAP-GlyΔ scs2/22Δ kar9Δ* mutant, where both MT capture-shrinkage and sliding were presumably defective. The number of cells, in which the anaphase spindle was misaligned in the mother cell compartment, was significantly enhanced for *CAP-GlyΔ scs2/22Δ kar9Δ* mutant compared with *scs2/22Δ kar9Δ* (47.2 vs. 35.7%, $n \geq 300$; *Figure 4F*). Additionally, *CAP-GlyΔ* mutation (capture-shrinkage disrupting) displayed severe synthetic viability defects with *scs2/22Δ* mutation (sliding disrupting) (*Figure 4G*). These data provide strong evidence indicating that loss of both dynein-mediated pulling activities could result in additive consequences to spindle positioning and cell viability.

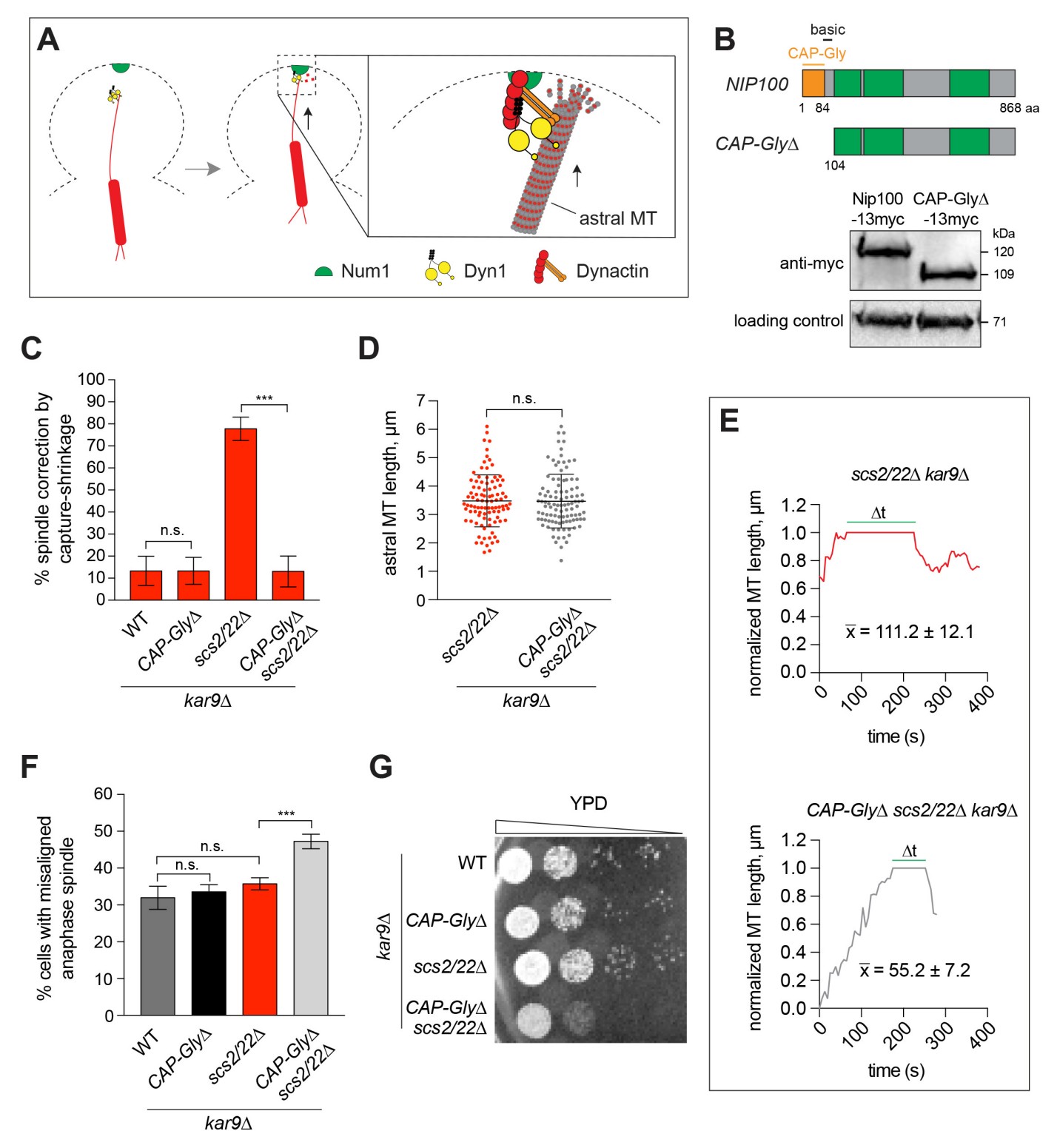

**Figure 4.** End-on capture-shrinkage of astral MT plus ends at the bud tip requires the CAP-Gly domain of Nip100/p150[Glued]. (**A**) Model showing MT tethering by Nip100/p150[Glued] during MT capture-shrinkage at the bud tip. (**B**) Schematic diagram of full-length and truncated Nip100/p150[Glued]. Western blot showing the expression levels of full-length Nip100-13myc in WT cells and CAP-GlyΔ-13myc in *scs2/22Δ* cells. (**C**) Spindle correction events that occurred via MT capture-shrinkage at the bud tip for each indicated strain during a 10-min movie (23 ≤ n ≤ 63 events per strain). Percentage was normalized as in *Figure 3B*. Error bars depict SEP. n.s., not statistically significant; ***p<0.0001 by one-way ANOVA test. (**D**) Quantification of astral MT length (mean ± SD, n ≥ 95 for each strain). n.s., not statistically significant by unpaired *t* test. (**E**) Representative traces
*Figure 4 continued on next page*

*Figure 4 continued*

showing the position of the MT plus end relative to the bud neck (position = 0) and the bud tip (position = 1) over time. x̄, average duration of contact (Δt) between the plus end and the bud tip (n ≥ 21). (**F**) Percentage of cells displaying a misaligned anaphase spindle phenotype for each indicated strain (n ≥ 216 cells per strain). Error bars indicate SEP. n.s., not statistically significant; ***p<0.0001 by one-way ANOVA test. (**G**) *CAP-GlyΔ* mutation displays synthetic growth defects with *scs2/22Δ* mutation. Serial dilutions of indicated strains in the *kar9Δ* background were spotted on rich medium plate and grown for 2 days at 30°C.

DOI: https://doi.org/10.7554/eLife.36745.019

The following figure supplement is available for figure 4:

**Figure supplement 1.** CAP-Gly domain is required for MT tethering during capture-shrinkage mechanism at the bud tip.

DOI: https://doi.org/10.7554/eLife.36745.020

## Lateral patches of Num1 along the bud cortex facilitate MT sliding

Our data suggest that changes in Num1 localization affect dynein pulling mechanism but not dynein pathway function. We next tested whether distribution of Num1 along the bud cortex could dictate the mechanism of dynein-mediated spindle positioning. To investigate this, we asked whether spindle correction via MT sliding could be rescued in the *scs2/22Δ* mutant if lateral patches of Num1 were restored along the bud cortex. We attached a CAAX motif to Num1-GFP and assessed dynein-dependent astral MT interaction with the bud cortex using a spindle correction assay.

Previous work showed that Num1-GFP-CAAX assembles functional cortical patches similar to those observed for Num1-GFP (*Tang et al., 2009*). We found that, unlike Num1-GFP patches (*Figure 1A*), Num1-GFP-CAAX patches were not affected by deletion of Scs2/22 and were distributed throughout the cell cortex (*Figure 5A*). Cortical foci of Dyn1-3mCherry and Jnm1-3mCherry were observed colocalizing with lateral Num1-GFP-CAAX patches in *scs2/22Δ* cells (*Figure 5—figure supplement 1A and B*). Interestingly, as lateral Num1 patches along the bud cortex were restored, we observed that dynein-dependent MT sliding became the primary mechanism for spindle correction in *NUM1-GFP-CAAX scs2/22Δ kar9Δ* cells: 80.0% (32 out of 40) of misaligned spindles were corrected by MT sliding mechanism compared with 22.2% (14 out of 63) when Num1-GFP was expressed in the same background (*Figure 5B* versus *Figure 3B*; *Video 7*). Consistent with the rescue of MT sliding mechanism, we found that the angles of interaction between the astral MT and the cortical surface for productive MT-cortex interactions (i.e. those followed by spindle correction) were significantly more oblique in *scs2/22Δ kar9Δ* cells expressing Num1-GFP-CAAX (54.0 ± 20°) compared with those expressing Num1-GFP (83.7 ± 12.8°) (*Figure 5C*). In control *kar9Δ* background, the angles for productive MT-cortex interactions for Num1-GFP-CAAX (40.3 ± 14.2) was similar to those observed for Num1-GFP (54.0 ± 16°), in agreement with the idea that Num1-GFP-CAAX forms functional patches like Num1-GFP. Additionally, considering all MT-Num1 interactions in the bud, we found that sliding was correlated with MT interacting with a Num1 patch located within the proximal three quarters of the bud cortex, whereas end-on pulling was correlated with MT interacting with a Num1 patch located within the distal quarter of the bud (*Figure 5D*).

Thus, when combined with our results in *Figure 3B*, wherein spindle correction is primarily mediated by MT capture-shrinkage mechanism when Num1 is limited to the bud tip, the aforementioned observations indicate that the distribution of Num1 along the bud cortex could govern the mechanism of dynein-mediated spindle positioning. Moreover, the rescue of MT

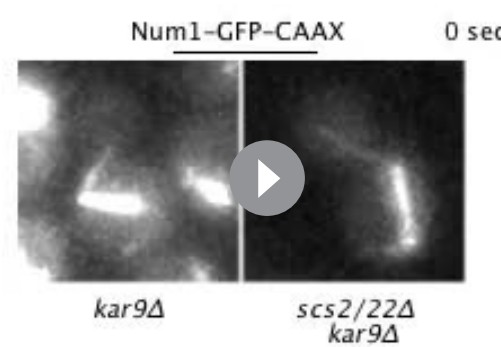

**Video 7.** Lateral Num1 patches restore astral MT sliding in *scs2/22Δ kar9Δ* cells. Time-lapse images of mRuby2-Tub1 in *kar9Δ* and *scs2/22Δ kar9Δ* cells expressing Num1-GFP-CAAX showing astral MT sliding along the bud cortex as the anaphase spindle translocated into the bud neck during its realignment. Each frame is a maximum intensity projection of 3 optical sections spaced 0.5 μm apart. Movie was captured at 10 s intervals for 6 min.

DOI: https://doi.org/10.7554/eLife.36745.025

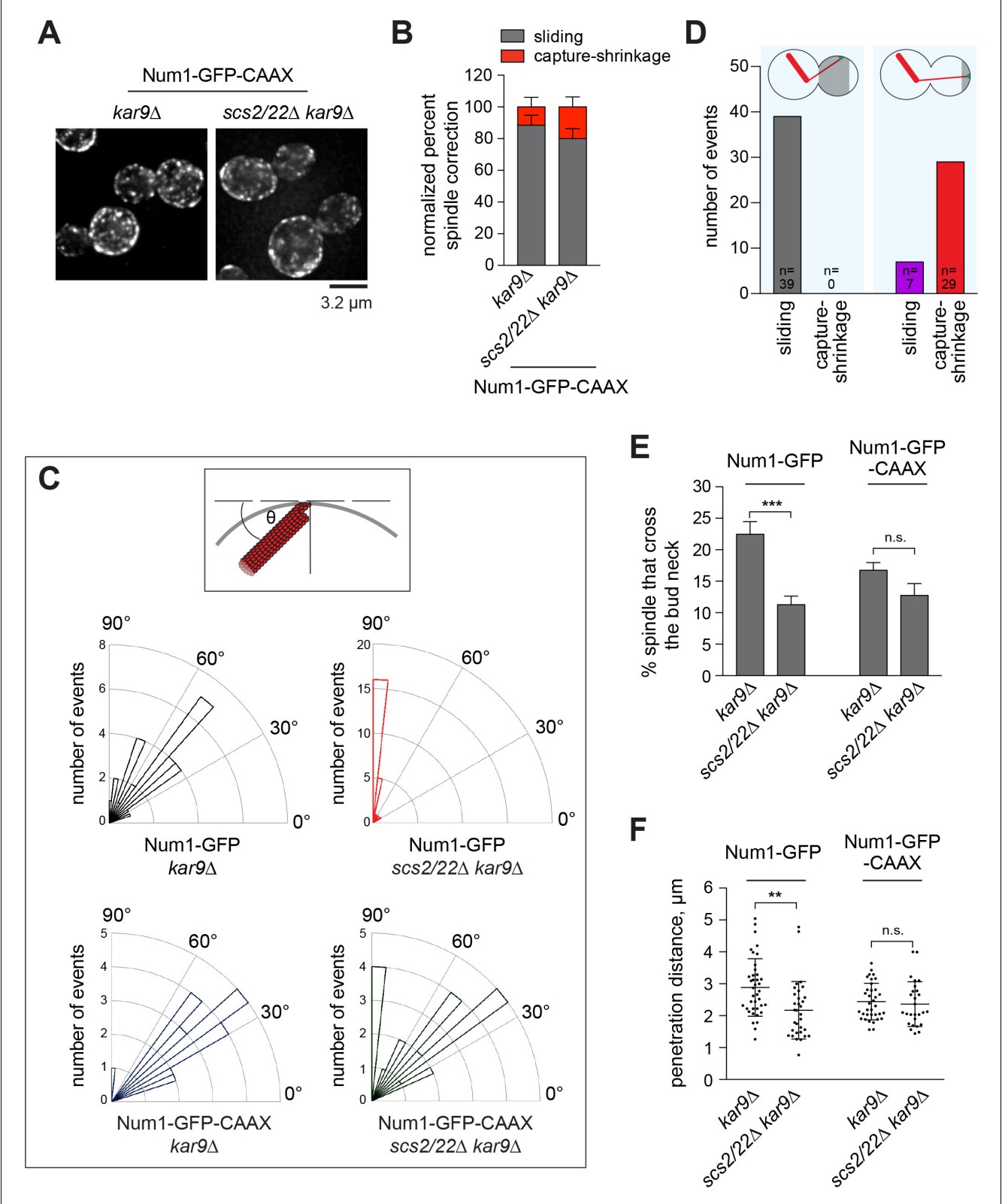

**Figure 5.** Lateral Num1 patches promote dynein-dependent MT sliding. (**A**) Maximum intensity projections of deconvolved wide-field images of *kar9Δ* and *scs2/22Δ kar9Δ* cells expressing Num1-GFP-CAAX. (**B**) Quantification of spindle correction mechanisms (n ≥ 26 cells per strain). Error bars indicate SEP. (**C**) *Top*, schematic showing the angle of interaction between the astral MT and the bud cortex. *Bottom*, rose histograms of the angle of interaction for each indicated strain (n ≥ 26 per strain). (**D**) Plot depicting the frequency of observing MT sliding or capture-shrinkage for MT-Num1

*Figure 5 continued on next page*

*Figure 5 continued*
interaction occurring within the proximal three quarters versus the distal quarter of the bud cortex. (E) Percentage of HU-arrested spindles that crossed the bud neck over the course of a 10 min movie. Error bars indicate SEP (n ≥ 298 per strain). n.s., not statistically significant. ***p<0.0001 by unpaired *t* test. (F) Penetration distance of HU-arrested spindles. Penetration distance is defined as the farthest distance traveled by the preanaphase spindle moving across the bud neck during a 10 min video. Error bars indicate SD (n ≥ 27). n.s., not statistically significant. **p<0.005 by unpaired *t* test.
DOI: https://doi.org/10.7554/eLife.36745.022
The following figure supplements are available for figure 5:

**Figure supplement 1.** Colocalization of cortical dynein and dynactin foci with Num1-GFP-CAAX patches.
DOI: https://doi.org/10.7554/eLife.36745.023
**Figure supplement 2.** Loss of Scs2/22 results in defective MT sliding along the bud cortex.
DOI: https://doi.org/10.7554/eLife.36745.024

sliding by Num1-GFP-CAAX in the absence of Scs2/22 suggests that the primary role of Scs2/22 in the dynein pathway is to distribute Num1 along the cell cortex to facilitate dynein-dependent MT sliding.

## MT sliding enhances efficiency of nuclear migration compared to MT capture-shrinkage

To examine the contribution of MT sliding to dynein pathway function in nuclear migration more closely, we quantitated spindle oscillation in HU-arrested cells, scoring for preanaphase spindle movements through the bud neck in a *kar9Δ* background. In *kar9Δ* cells, these movements coincided with lateral sliding of an astral MT along the cell cortex (*Moore et al., 2009*). Lateral distribution of Num1 along the cortex might be necessary to promote efficient spindle movement across the bud neck. Compared with *kar9Δ*, *scs2/22Δ kar9Δ* mutant lacking lateral Num1 patches exhibited a significantly lower number of cells in which the preanaphase spindle moved from the mother cell compartment through the bud neck (22.5 vs. 11.2%, p<0.0001; *Figure 5E*, left). Moreover, in cells where the spindle was able to penetrate the bud neck, it moved for a significantly shorter distance (*Figure 5F*, left). The observed differences could not be attributed to changes in astral MT dynamics in *scs2/22Δ kar9Δ* mutant (*Table 2*). However, we found that Num1-GFP-CAAX, which restored lateral Num1 patches and lateral MT sliding in *scs2/22Δ kar9Δ* (*Figure 5A and B*), rescued the frequency of spindle movement across the bud neck to a level similar to that observed in *kar9Δ* (12.8 versus 16.7%, p=0.096; *Figure 5E*, right). Num1-GFP-CAAX also rescued the spindle penetration distance to a *kar9Δ* level (*Figure 5F*, right), consistent with a role for lateral Num1 patches and MT sliding in increasing the efficiency of nuclear migration. These analyses uncovered a compromised dynein function in the *scs2/22Δ* cells, albeit without resulting in a spindle misorientation phenotype (*Figure 2F*).

## Num1 localization at the bud tip in *scs2/22Δ* cells depends on the formin Bni1

We next investigated how Num1 is targeted to the bud tip in *scs2/22Δ* cells. Our results thus far indicate that Num1 redistributes to the bud tip when cortical ER tethering (along the cell periphery) is disrupted by deletion of Scs2/22. Interestingly, a previous study using overexpressed epitope-tagged proteins showed that Num1 co-precipitated with the formin Bni1 (*Farkasovsky and Küntzel, 2001*), a polarisome component that nucleates actin cables in the bud (*Evangelista et al., 2002*; *Sagot et al., 2002*), suggesting that Bni1 and/or actin may play a role in Num1 targeting to the bud

**Table 2.** Parameters of MT dynamics for free astral MTs (i.e. unattached to cortex) in *kar9Δ* and *scs2/22Δ kar9Δ* mutants.

|  | *kar9Δ* | *scs2/22Δ kar9Δ* |
| --- | --- | --- |
| Growth rate (μm/min) | 1.38 ± 0.12 (n = 70) | 1.07 ± 0.07 (n = 61) |
| Shrinkage rate (μm/min) | 1.63 ± 0.12 (n = 59) | 1.70 ± 0.14 (n = 67) |
| Catastrophe frequency (event/min) | 0.56 ± 0.07 (n = 19) | 0.42 ± 0.09 (n = 17) |
| Rescue frequency (event/min) | 0.59 ± 0.08 (n = 18) | 0.40 ± 0.06 (n = 16) |

DOI: https://doi.org/10.7554/eLife.36745.026

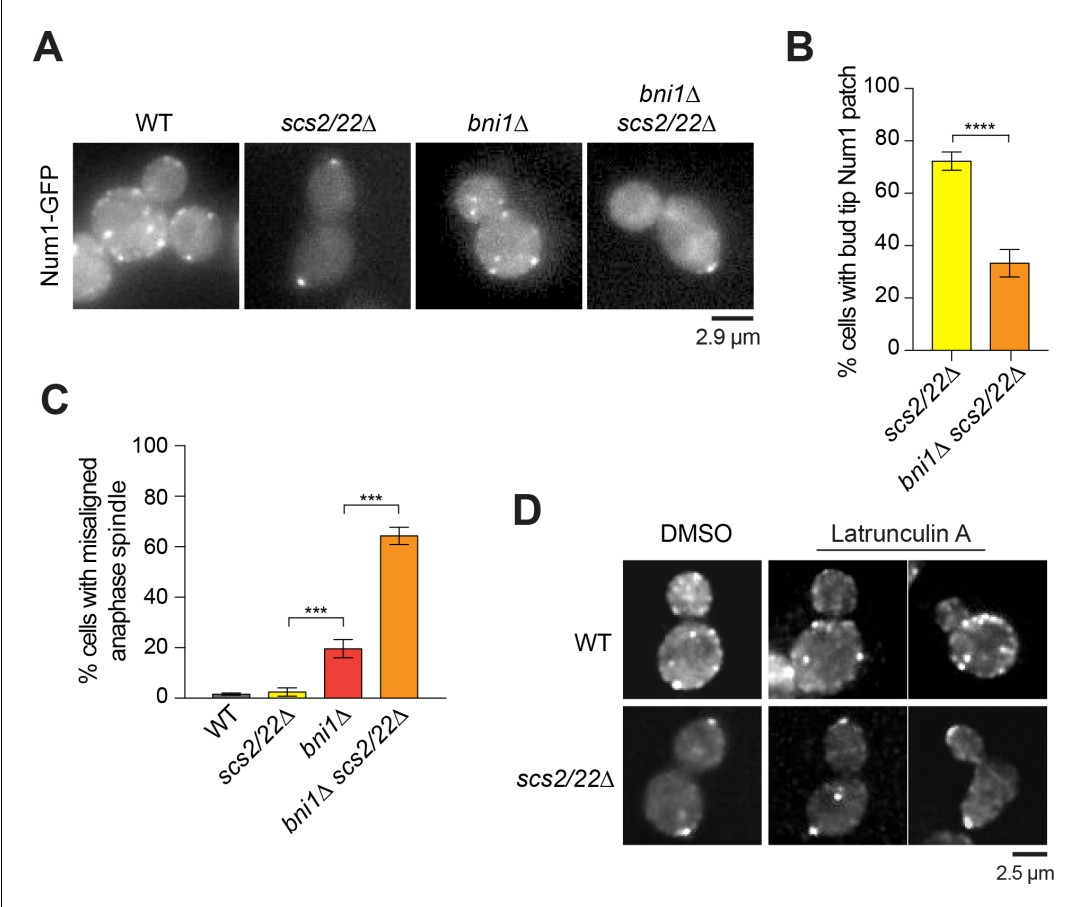

**Figure 6.** Num1 localization at the bud tip in *scs2/22Δ* requires Bni1. (A) Wide-field images of Num1-GFP in WT, *scs2/22Δ*, *bni1Δ*, and *bni1Δ scs2/22Δ* cells. Each image is a maximum intensity projection of 7 optical sections spaced 0.5 μm apart. (B) Percentage of cells with Num1-GFP patch at the bud tip is decreased in *bni1Δ scs2/22Δ* mutant relative to *scs2/22Δ* mutant (n ≥ 81 cells per strain). Error bars indicate SEP. ****p<0.0001 by unpaired *t* test. (C) Percentage of misaligned anaphase spindle for WT, *scs2/22Δ*, *bni1Δ*, and *bni1Δ scs2/22Δ* cells (n ≥ 83 per strain). Error bars indicate SEP. ***p<0.0001 by one-way ANOVA test. (D) Num1-GFP localization in WT and *scs2/22Δ* cells treated with DMSO or 200 μM latrunculin A for 20 min.
DOI: https://doi.org/10.7554/eLife.36745.027

The following figure supplement is available for figure 6:

**Figure supplement 1.** Control experiment showing F-actin disassembly by latrunculin A.
DOI: https://doi.org/10.7554/eLife.36745.028

tip. We found that the percentage of *scs2/22Δ* cells exhibiting a Num1-GFP patch at the bud tip was significantly decreased upon deletion of Bni1 (72.3 to 33.3%; *Figure 6A and B*; *Video 8*). The reduction in Num1 bud tip localization was accompanied by a striking spindle misalignment phenotype (*Figure 6C*): 64.3% of *bni1Δ scs2/22Δ* cells exhibited a misaligned anaphase spindle compared with 2.5% of *scs2/22Δ* and 1.5% of WT cells. Notably, the levels of the spindle misalignment phenotype in *bni1Δ scs2/22Δ* cells were enhanced significantly (by ~3.3 fold) relative to those observed in *bni1Δ* single mutant (*Figure 6C*), indicating a synergistic defect between *bni1Δ* and *scs2/22Δ* in anaphase spindle alignment. Furthermore, we found that depolymerization of F-actin using latrunculin A did not perturb Num1 localization at the bud tip (*Figure 6D*), even though F-actin was completely disassembled, as judged by rhodamine-phalloidin staining (*Figure 6—figure supplement 1*). These data show that maintenance of Num1-GFP patches is independent of F-actin in *scs2/22Δ* cells, consistent with a previous study in WT cells (*Heil-Chapdelaine et al., 2000*). Together, these results support that Bni1 itself, rather than its actin nucleation activity, is required for Num1 localization and function at the bud tip. Alternatively, Bni1 might be required early in the cell cycle to establish a binding site for Num1 attachment later in the cell cycle.

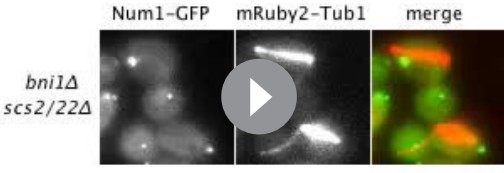

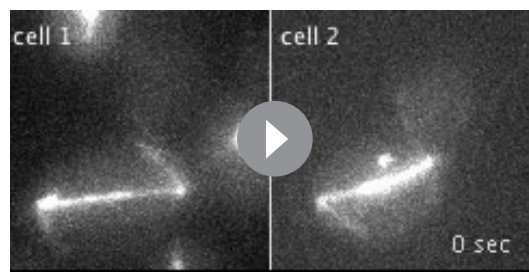

**Video 8.** Bni1 is required for Num1 localization at the bud tip in *scs2/22Δ* cells. Time-lapse images of Num1-GFP and mRuby2-Tub1 in the *bni1Δ scs2/22Δ* background. Each frame is a maximum intensity projection of 5 optical sections spaced 0.5 μm apart. Movie was captured at 10 s intervals.

DOI: https://doi.org/10.7554/eLife.36745.029

**Video 9.** Astral MT stops sliding upon reaching the bud tip. Time-lapse images of mRuby2-Tub1 in *kar9Δ* background showing examples of MT sliding events being terminated at the bud tip. Each frame is a maximum intensity projection of 7 optical sections spaced 0.5 μm apart. Movie was captured at 5 s intervals.

DOI: https://doi.org/10.7554/eLife.36745.031

## Dynein-dependent MT capture-shrinkage regulates MT sliding in the bud

Given the dual modes of dynein pulling mechanisms, we wondered whether they might function together to regulate spindle movement into the bud neck in WT cells. Interestingly, MT sliding movies from previous studies showed that productive sliding events in WT cells were often initiated along the lateral bud cortex and terminated when the plus end of the sliding MT encountered the bud tip (see Video 1 in *Lee et al., 2003*). Additionally, *Yeh et al. (2000)* reported that astral MTs frequently undergo depolymerization at the bud tip after a dynein-dependent sliding event that pulled the anaphase spindle into the bud neck (see Figure 7 in *Yeh et al., 2000*).

To interrogate this further, we examined MT behavior during the end of MT sliding events in a *kar9Δ*, but an otherwise WT, background. In 16 out of 33 (~49%) spindle correction events that began as sliding, we observed that the astral MT stopped sliding upon reaching the bud tip (*Figure 7A* and *Video 9*). Significantly, as shown in *Videos 10* and *11*, colocalization with Num1-

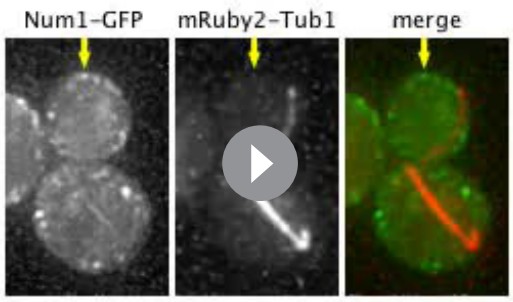

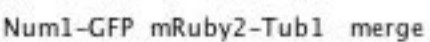

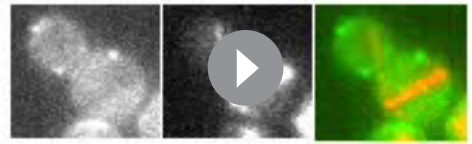

**Video 10.** Astral MT stops sliding when the plus end encounters a Num1 cluster at the bud tip. Time-lapse images of Num1-GFP and mRuby2-Tub1 in *kar9Δ* showing an example of MT sliding being halted when the plus end of the sliding MT reaches a Num1 cluster at the bud tip. Arrow indicates the position of the Num1 cluster (visible in the initial frames before photobleaching) where stoppage of MT sliding occurs. Each frame is a maximum intensity projection of deconvolved wide-field images with five optical sections spaced 0.5 μm apart. Movie was captured at 15 s intervals. Note that mRuby2 fluorescence (mRuby2-Tub1) appears to crossover slightly into the GFP channel (Num1-GFP).

DOI: https://doi.org/10.7554/eLife.36745.032

**Video 11.** A second example of MT sliding stoppage occurring when the plus end encounters a Num1 patch at the bud tip. Time-lapse images of Num1-GFP and mRuby2-Tub1 in *kar9Δ ist2Δ* background showing a clear example of MT sliding being halted when the plus end of the sliding MT reaches a Num1 patch at the bud tip. *KAR9* and *IST2* deletions did not affect Num1 localization in WT cells (data not shown). Each frame is a maximum intensity projection of 3 optical sections spaced 0.5 μm apart. Movie was captured at 10 s intervals.

DOI: https://doi.org/10.7554/eLife.36745.033

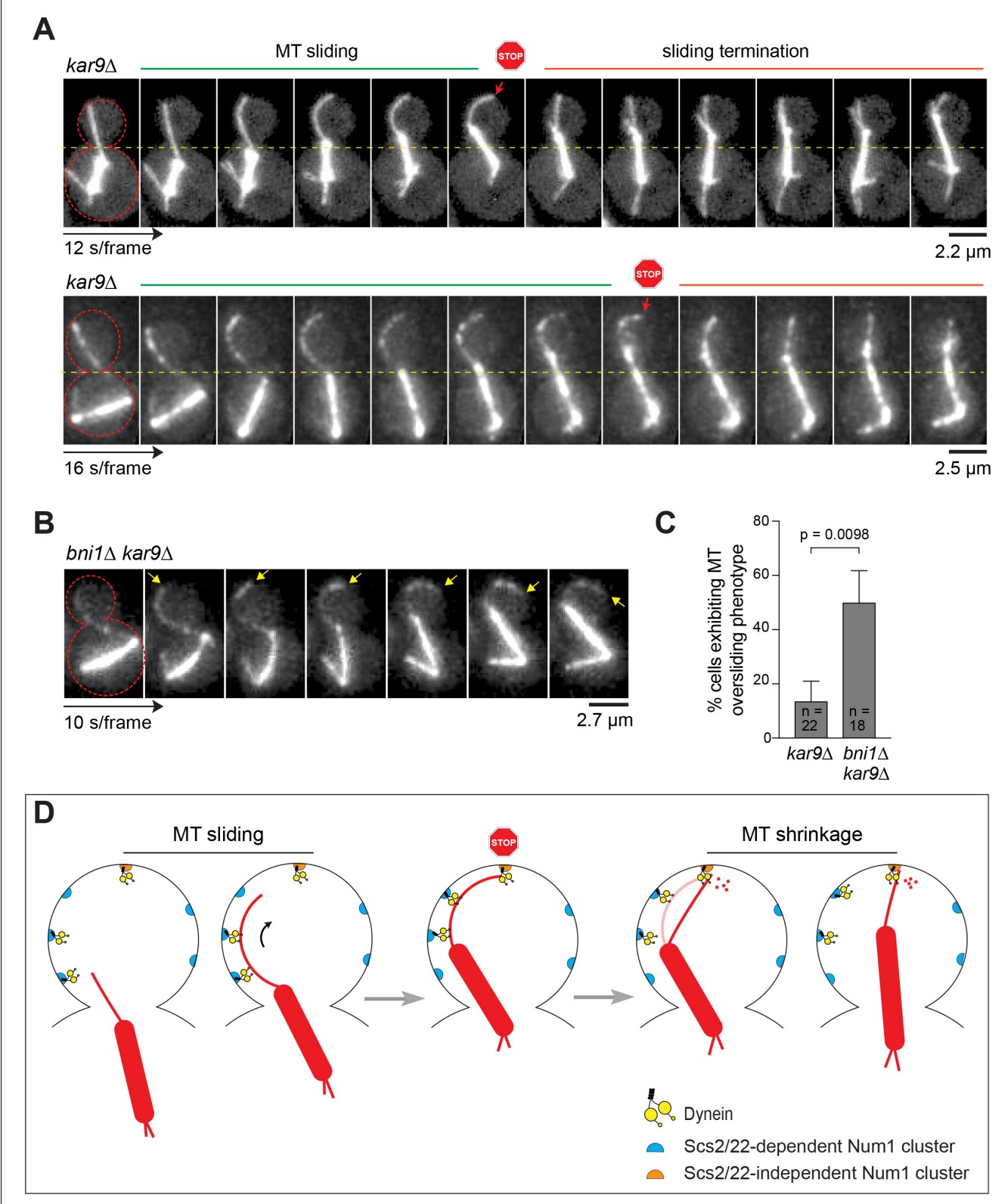

**Figure 7.** Proposed mechanism for Num1 regulation of dynein pulling force along the bud cortex. (**A**) Movie frames of representative MT sliding events in *kar9Δ* background. *Top*, confocal; *bottom*, deconvolved wide-field. Each frame is a maximum intensity projection of 7–9 optical sections spaced 0.5 μm apart. Red arrows indicate the transition from side-on to end-on conformation at the MT capture site. (**B**) Movie frames of MT oversliding phenotype in the *bni1Δ kar9Δ* background. Each frame is a maximum intensity projection of wide-field images with five optical sections spaced 0.5 μm *Figure 7 continued on next page*

*Figure 7 continued*

apart. Yellow arrows indicate the position of the MT plus end. (C) Frequency of observing MT oversliding phenotype during spindle correction in *kar9Δ* and *bni1Δ kar9Δ* cells. Error bars indicate SEP. (D) Model showing regulation of dynein pulling mechanism by two spatially distributed populations of Num1. (Step 1) ER-associated Num1 mediates lateral dynein-dependent MT-cortex interaction, pulling the astral MT along the bud cortex toward the bud tip. (Step 2) MT plus end is captured by dynein anchored at the bud tip by Scs2/22-independent Num1, stopping MT sliding. (Step 3) The motor activity of dynein induces MT depolymerization at the bud tip, causing the plastered astral MT to shorten and straighten out. As the MT shrinks, the spindle is moved closer to the bud tip, further aligning the spindle along the mother-bud axis. For clarity, plus end-targeted dynein and cortical ER are omitted from the diagram. See *Videos 9–12* and text for further discussion.

DOI: https://doi.org/10.7554/eLife.36745.030

GFP indicated that stoppage of sliding occurred when the MT plus end encountered a Num1 patch or a cluster of Num1 at the bud tip. In all cases, at the end of sliding, the cortex-plastered astral MT could be seen changing into a straight conformation, with the plus end remaining attached to the bud tip. Because the straightening event did not push the spindle back toward the mother cell, the transition (from a plastered/bent conformation to a straight conformation) suggested that the captured plus end was undergoing depolymerization at the bud tip. We observed that, in majority of the cases (10 out of 16; 63%), the attached MT continued to shorten and shrink, causing the spindle to move closer to the bud tip, further aligning the spindle along the mother-bud axis. In the remaining cases (6 out of 16; 37%), straightening was followed by the immediate release of the astral MT from the bud tip. Intriguingly, in *kar9Δ* cells lacking the formin Bni1, which is required for Num1 localization to the bud tip (*Figure 6A and B*) (*Farkasovsky and Küntzel, 2001*), we observed a significant increase in the frequency of finding astral MT sliding that went past the bud tip instead of stopping upon reaching the bud tip (*Figure 7B and C*; *Video 12*). Together these results suggest that dynein-dependent MT capture-shrinkage at the bud tip has a role in regulating spindle movement in WT cells.

## Discussion

Our studies show how changes in cortical distribution of Num1 can dramatically alter dynein-dependent spindle pulling mechanism. When Num1 is restricted to the bud tip, as in the case for *scs2/22Δ* cells, dynein generates pulling forces predominantly via the MT capture-shrinkage mechanism. In this configuration, anchored dynein is geometrically limited to interact with the very end of the astral MT. Additionally, our data suggest that, through CAP-Gly domain of Nip100/p150[Glued], dynactin may act as a cortical linker to maintain the connection between the shrinking MT end and the cortex, enabling transmission of MT depolymerization into pulling force generated by the dynein motor at the bud tip. However, when Num1 is distributed along the bud cortex, as in the case for WT and *NUM1-GFP-CAAX scs2/22Δ kar9Δ* cells, dynein generates pulling forces primarily via the MT sliding mechanism. In this configuration, anchored dynein can pull on the spindle by moving laterally along the MT lattice. This type of lateral cortical contact is facilitated by a larger surface for dynein-MT interaction, explaining why lateral Num1 patches are more efficient in promoting spindle movement across the bud neck (*Figure 5E and F*). Interestingly, our studies also show how stoppage of MT sliding is coupled with MT capture and shrinkage at the bud tip, providing a mechanism by which dynein-dependent cortical pulling is spatially regulated in the bud. We show that the population of Num1 at the bud tip depends on Bni1. We propose that MT capture-shrinkage at the bud tip functions as a brake for MT sliding (*Figure 7D*). This function may be

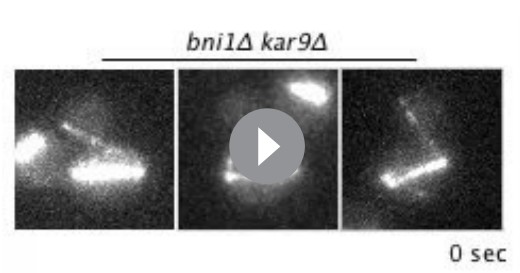

**Video 12.** Astral MT slides past the bud tip instead of stopping upon reaching the bud tip in *bni1Δ kar9Δ* cells. Time-lapse images of mRuby2-Tub1 showing three examples of MT oversliding phenotype at the bud tip in the *bni1Δ kar9Δ* background. Each frame is a maximum intensity projection of 5 optical sections spaced 0.5 μm apart. Movie was captured at 10 s intervals.

DOI: https://doi.org/10.7554/eLife.36745.034

important to prevent oversliding of MT beyond the bud tip, thereby ensuring that the spindle is correctly positioned across the bud neck.

Our results suggest that cortical ER has a facilitative role in the dynein pathway. In wild-type yeast cells, the cortical ER is consisted of a network of sheets and tubules that tightly associate with the plasma membrane (*Pichler et al., 2001*; *Prinz et al., 2000*; *West et al., 2011*). The cortical ER and the PM make extensive contacts along the periphery of the cell, forming structures called ER-PM junctions (*Prinz, 2014*; *Stefan et al., 2013*), which have been implicated in various cellular processes, including phosphoinositide signaling (*Stefan et al., 2011*), sterol lipid transport (*Schulz et al., 2009*), as well as maintenance of ER morphology and regulation of the unfolded protein response in the ER (*Manford et al., 2012*). At least three families of integral ER proteins – Scs2/22, Ist2, and Tcb1/2/3 – function to tether the cortical ER to the PM (*Eisenberg-Bord et al., 2016*; *Manford et al., 2012*). Among them, Scs2/22 appear to be the most important (*Loewen et al., 2007*; *Manford et al., 2012*). They contain a single transmembrane domain and a cytoplasmic MSP (major sperm protein) domain that can bind directly to PI lipids (*Kagiwada and Hashimoto, 2007*) or a FFAT (diphenylalanine in an acidic tract) motif found in lipid transfer proteins (*Loewen et al., 2003*). Interestingly, a recent study shows that Num1 has a putative FFAT motif in its N-terminal region that targets Num1 to the cortical ER by binding to the MSP domain of Scs2 (*Chao et al., 2014*). This raises the possibility that cortical ER-PM junctions may play a role in anchoring Num1 (and therefore dynein) for mediating movement of the nucleus into the bud neck. Our study here provides evidence to support a specific role in facilitating MT sliding. First, in WT cells, Num1 localizes to numerous dim patches throughout the cell cortex. In particular, dim patches are observed along the lateral cortex of medium and large buds (*Figure 1A*), where dynein-dependent MT sliding is thought to occur during movement of the nucleus into the bud neck. We speculate that these dim patches represent Num1 molecules that are anchored at the ER-PM contact sites, since loss of Scs2 (*Figure 1—figure supplement 1B*) or Scs2/22 (*Figure 1A*) resulted in a severe loss of dim patches. Second, the number of MT sliding events that occur along the bud cortex is significantly diminished by loss of Scs2/22 (*Figure 5—figure supplement 2A*). Although some sliding events remained, they appeared to be less effective in moving the spindle (*Figure 5—figure supplement 2B*). It is possible that the remaining sliding events were mediated by Num1 that binds to the PM via its C-terminal PH domain (*Yu et al., 2004*) or to specific membrane on the mitochondrial surface via its N-terminal CC domain (*Ping et al., 2016*). However, the reduction in the spindle penetration distance observed in the absence of Scs2/22 (*Figure 5F* and *Figure 5—figure supplement 2B*) is consistent with the idea that Num1 attachment to ER-PM junctions can provide a stronger resistive force, which may be needed to firmly anchor dynein for efficient pulling of the spindle from the cell cortex.

Our study demonstrates for the first time that dynein pulling forces are spatially mediated and regulated by two differential populations of Num1 patches in the WT yeast buds. The first population, namely the Scs2/22-dependent lateral patches (as discussed above), appears to initiate and facilitate MT sliding along the bud cortex. The second population, which localizes to the bud tips, terminates MT sliding by capturing MT plus end and inducing MT catastrophe. Consistent with the idea that dynein pulling activity is tightly regulated, previous work shows that the ER diffusion barrier also functions at the bud neck to confine Num1 to the mother compartment until M phase (*Chao et al., 2014*). The diffusion barrier appears to be important for regulating the start of dynein pulling activity, by preventing premature localization of Num1 into the bud compartment. Our study now provides an additional level of control for modulating dynein pulling activity, which appears to be important for ending the spindle movement once it is started in the bud (*Figure 7D*). Intriguingly, we did not observe premature accumulation of Num1 to the tips of small buds in *scs2/22Δ* cells (*Figure 1—figure supplement 4*), where the ER diffusion barrier is presumably disrupted, suggesting that additional components might be required to recruit Num1 to the bud tip in a timely manner.

How Num1 switches the dynein motor from a side-on motor to an end-on motor remains an open question at this point. Our data rule out Scs2/22 as being required for motor activity, given that MT sliding can be rescued in the absence of Scs2/22 by a CAAX-targeted Num1. Since the mechanism of membrane attachment for Num1 is likely to be different between the lateral cortex and the bud tip, it will be interesting for future investigations to examine whether cortical stiffness could play a role in regulating the switching of cortical dynein pulling mechanisms.

# Materials and methods

**Key resources table**

| Reagent type (species) or resource | Designation | Source or reference | Identifiers | Additional information |
|---|---|---|---|---|
| Antibody | anti-c-Myc (mouse monoclonal) | BioLegend | BioLegend:626802; RRID:AB_2148451 | (1:250 or 500 or 1000) |
| Antibody | Rabbit IgG | GenScript | GenScript:A01008; RRID:AB_2732863 | (1:5000) |
| Antibody | HRP goat anti-mouse IgG | BioLegend | BioLegend:405306; RRID:AB_315009 | (1:10000) |
| Antibody | HRP goat anti-rabbit IgG | Jackson Immuno Research Labs | Jackson ImmunoResearch Labs: 111-035-144; RRID:AB_2307391 | (1:10000) |
| Antibody | anti-Sac1 (rabbit polyclonal) | Charles Barlowe Lab | NA | (1:2000) |
| Gene (*Saccharomyces cerevisiae*) | NUM1 | NA | SGD:S000002557 | Systematic name: YDR150W |
| Gene (*S. cerevisiae*) | SCS2 | NA | SGD:S000000922 | Systematic name: YER120W |
| Gene (*S. cerevisiae*) | SCS22 | NA | SGD:S000007228 | Systematic name: YBL091C-A |
| Gene (*S. cerevisiae*) | DYN1 | NA | SGD:S000001762 | Systematic name: YKR054C |
| Gene (*S. cerevisiae*) | NIP100 | NA | SGD:S000006095 | Systematic name: YPL174C |
| Gene (*S. cerevisiae*) | JNM1 | NA | SGD:S000004908 | Systematic name: YMR294W |
| Gene (*S. cerevisiae*) | ARP1 | NA | SGD:S000001171 | Systematic name: YHR129C |
| Gene (*S. cerevisiae*) | KAR9 | NA | SGD:S000006190 | Systematic name: YPL269W |
| Gene (*S. cerevisiae*) | BNI1 | NA | SGD:S000005215 | Systematic name: YNL271C |
| Gene (*S. cerevisiae*) | MMR1 | NA | SGD:S000004180 | Systematic name: YLR190W |
| Gene (*S. cerevisiae*) | GEM1 | NA | SGD:S000000046 | Systematic name: YAL048C |
| Gene (*S. cerevisiae*) | PAC1 | NA | SGD:S000005795 | Systematic name: YOR269W |
| Gene (*S. cerevisiae*) | CIN8 | NA | SGD:S000000787 | Systematic name: YEL061C |
| Gene (*S. cerevisiae*) | KIP3 | NA | SGD:S000003184 | Systematic name: YGL216W |
| Gene (*S. cerevisiae*) | KAR3 | NA | SGD:S000006345 | Systematic name: YPR141C |
| Chemical compound, drug | Hydroxyurea | Thermo Fisher Scientific | Thermo Fisher Scientific: AC151680250 | (200 mM) |
| Chemical compound, drug | Latrunculin A | Millipore Sigma | Millipore Sigma:L5163 | (200 µM) |
| Chemical compound, drug | Hygromycin B | Gold Biotechnology | Gold Biotechnology: H-270–5 | (400 µg/ml) |
| Chemical compound, drug | G-418 sulfate | Gold Biotechnology | Gold Biotechnology: G-418–10 | (400 µg/ml) |
| Software, algorithm | NIS-Elements | Nikon | RRID:SCR_014329 | |
| Software, algorithm | ImageJ | NIH | RRID:SCR_003070 | |

*Continued on next page*

*Continued*

| Reagent type (species) or resource | Designation | Source or reference | Identifiers | Additional information |
|---|---|---|---|---|
| Software, algorithm | Prism 7 | GraphPad | RRID:SCR_002798 | |
| Software, algorithm | Illustrator | Adobe | RRID:SCR_010279 | |
| Software, algorithm | MATLAB | MathWorks | RRID:SCR_001622 | |
| Other | Rhodamine-phalloidin | Cytoskeleton | Cytoskeleton:PHDR1 | (1.4 µM) |
| Other | Protease inhibitor cocktail tablet | Millipore Sigma | Millipore Sigma:5892970001 | (1 tablet per 7 ml) |

## Media and strain construction

All strains used in this study are listed in *Supplementary file 1* and were derived from the genetic background of WT strains YWL36 and YWL37 (*Vorvis et al., 2008*) or the protease-deficient strain BJ5457 (*Jones, 1990*). Strains were generated by standard genetic crosses or by PCR product-mediated transformations (*Longtine et al., 1998*). Diploids resulted from each cross were sporulated and tetrad dissected and the progeny were then examined by marker analysis. Transformations were performed using lithium acetate protocol (*Knop et al., 1999*). Transformants were purified twice by streaking to single colonies on selective media plates. Proper deletion or insertion of fluorescent protein tagging cassette at the genomic locus was confirmed by diagnostic PCR and fluorescence microscopy. All fluorescent protein tagging was done at the chromosomal locus and imaging was performed using live cells unless stated otherwise. At least two independent transformants were chosen from each disruption or tagging procedure for subsequent experiments.

To label MTs, strains were transformed with HindIII-digested *HIS3p::mCherry-TUB1::LEU2* (*Zhu et al., 2017*) or BsaBI-digested *HIS3p:mRuby2-TUB1+3'UTR::URA3* and *HIS3p:mRuby2-TUB1 +3'UTR::LEU2* (*Markus et al., 2015*), or undigested *GFP-TUB1::LEU2* (*Song and Lee, 2001*). Transformants were screened and selected by fluorescence microscopy. To label endoplasmic reticulum, we constructed a plasmid expressing EGFP-HDEL via Gibson assembly reaction (*Gibson et al., 2009*) for integration at the *URA3* locus. Briefly, PCR products containing the *TEF1* promoter (*TEF1p*, 459 bp upstream of *TEF1* start codon), ER-targeting signal sequence (SS, 126 bp of the 5' end of *KAR2*), 3XGlyAla linker, and EGFP-HDEL sequence (amplified from pFA6a-GFP(S65T)-TRP1 plasmid (*Longtine et al., 1998*) with HDEL sequence built into the reverse PCR primer) were assembled together and cloned into BamHI and NotI digested pRS315, a *LEU2*-containing vector (*Sikorski and Hieter, 1989*). Next, we subcloned *TEF1p-SS-3xGlyAla-GFP-HDEL* as a HindIII-SacI fragment into pRS306, a *URA3*-containing vector, generating pRS306-*TEF1p-SS-3xGlyAla-GFP-HDEL::URA3*. Strains were transformed with StuI-linearized pRS306-*TEF1p-SS-3xGlyAla-GFP-HDEL:: URA3* for integration into *URA3* locus. Ura+ transformants were selected, colony purified, and screened by fluorescence microscopy.

To generate in-frame deletion of the CAP-Gly and basic domain of Nip100 at the endogenous chromosomal locus, we used the two-step approach for constructing unmarked genomic mutagenesis (*Gray et al., 2005*). Briefly, the *URA3* marker was amplified from pRS306 with primers containing sequences flanking the targeted region of Nip100 (amino acid 2–103). We verified the substitution of the targeted sequence with *URA3* by diagnostic PCR from the genomic DNA. The resulting strain was transformed with a second PCR product containing an in-frame fusion of the sequences flanking the targeted region (60 bp on one side of the *URA3* insertion and 1209 bp on the other side), amplified from WT genomic DNA, along with a carrier plasmid containing the *LEU2* marker (pRS315). Transformants were replica plated to 5-fluoroorotic acid (5-FOA) plates to select for the removal of *URA3*. Deletion of the targeted sequence was confirmed by diagnostic colony PCR and DNA sequencing.

## Image acquisition and analysis

Confocal images and fluorescence recovery after photobleaching (FRAP) experiments were acquired using a 1.4 NA 60X oil immersion objective on a Nikon A1R confocal microscope equipped with a LU-NB laser launch system housed in the IALS Nikon Center of Excellence microscopy facility at UMass Amherst. Pinhole size was set to 0.7 airy unit. To FRAP, we bleached for 3 s using 488 nm

laser at 5% laser power. After photobleaching, single focal plane images were acquired every 30 s at 0.3% laser power. Wide-field fluorescence images were acquired using either a 1.45 NA 100X objective on a Nikon 80i upright microscope equipped with piezo Z control (Physik Instrumente) and a cooled electron-multiplying charged-coupled device (EMCCD) camera (Cascade II; Photometrics) or a 1.49 NA 100X objective on a Nikon TiE inverted microscope system equipped with a laser launch (405/488/561/640 nm; LUN4; Nikon) and a EMCCD camera (iXon 888; Andor). Filter cube sets (31000 v2, 49002, 49008, and TRF89901; Chroma) were used for imaging DAPI, GFP and mRuby2/mCherry fluorescence. All three microscope systems were controlled by NIS-Elements software (Nikon). Yeast strains were grown to mid-log phase in synthetic-defined media (Sunrise Science Products, CA) at 30°C and mounted on 1.7% agarose pad for imaging. All images were acquired at room temperature. For three-dimensional reconstruction of Num1-GFP localization, we acquired up to 25 optical confocal sections spaced 0.3 µm apart encompassing the entire thickness of cells. Image stacks were deconvolved where indicated using the 3D Deconvolution tool in NIS-Elements software. To minimize phototoxicity to cells and photobleaching during time-lapse imaging, we acquired frames at intervals as indicated in the videos and with up to seven optical sections spaced 0.5 µm apart.

To quantify the number of cortical Num1 patches per cell, we used the analyze particle tool in ImageJ to determine the number of Num1-GFP foci from maximum intensity projection of confocal Z-stack images. Cortical patches were defined as foci having intensities above the threshold set by the average background intensity measured within the cytoplasmic area. To determine the spatial distribution of Num1 patches, we used the multipoint tool in ImageJ to determine the x-y coordinates of individual Num1-GFP patches relative to the bud neck from a maximum intensity projection of Z-stack images. The position of each patch was then normalized along the y-axis, with the distance from the bud neck to the bud tip as 0 to −1, and the distance from the bud neck to the mother apex as 0 to 1 (see diagram in *Figure 1C*). To measure the intensity of Num1 patches, we used the circle tool in ImageJ to encompass individual Num1-GFP foci from maximum intensity projection of Z-stack images. To subtract the background intensity from each measurement, we moved the circle tool from the patch to a nearby cytoplasmic area within the same cell. To quantify colocalization of Num1-GFP with Scs2-mRuby2, we used the Colocalization tool in NIS-Elements software. Pearson's correlation coefficients between Num1-GFP and Scs2-mRuby2 were calculated for individual Num1 patches in two-color deconvolved wide-field images of Num1-GFP and Scs2-mRuby2. Time-lapse videos displaying XY and XZ views were generated using XYZ projection tool in ImageJ. Kymographs were generated using the MultipleKymograph plugin for ImageJ. We used the mTrack plugin for ImageJ to track the MT plus end in cells expressing mRuby2-Tub1. The position of the plus end relative to the bud neck was tracked over time in movies acquired with 5 s intervals. The duration of plus end attachment at the bud tip (Δt) was scored as the length of time between the plus end making contact with the bud tip and the time it depolymerizes away from the bud tip.

For cold spindle misorientation assay, mid-log cultures expressing fluorescently-labeled tubulin were grown in YPD and then shifted to 16°C for 15 hr before imaging. For cold nuclear segregation assay, mid-log cultures were grown in YPD and then shifted to 16°C for 15 hr, fixed with 70% ethanol and stained with DAPI. For spindle oscillation assay, strains expressing mRuby2-Tub1 were grown to mid-log and arrested with 200 mM hydroxyurea for 1–1.5 hr before imaging, as described (*Moore et al., 2009*; *Tang et al., 2012*). The velocity of spindle movement was defined as ΔD/ΔT, in which ΔD was the distance the spindle traveled in a continuous bud-directed movement, and ΔT was the time for the movement. For spindle correction assay, we scored for misoriented anaphase spindles that moved into the bud neck and became aligned along the mother-bud axis during a 10 min movie in the *kar9Δ* background. Spindle correction was scored as mediated by 'sliding mechanism' if the astral MT displayed lateral association with the bud cortex while the spindle moved into the bud neck during its realignment, and by 'capture-shrinkage mechanism' if the astral MT exhibited end-on interaction with the bud tip followed by depolymerization of the astral MT concomitant with spindle movement into the bud neck. MT growth rate, shortening rate, catastrophe frequency, and rescue frequency were measured as described (*Gupta et al., 2006*). The angle of interaction between astral MT and the bud tip was measured using the angle tool in ImageJ. Rose histograms were plotted using MATLAB. For growth assays, strains were grown to mid-log phase in YPD media, then ten-fold serial dilutions were spotted on YPD plates and grown at 30°C for 2 days. To depolymerize F-actin, cells were grown to early log phase, collected by centrifugation, and resuspended in synthetic-

defined medium containing 200 µM latrunculin A or 0.5% DMSO for 20 min before imaging. To verify loss of F-actin, cells were fixed and stained with rhodamine-phalloidin as previously described (*Waddle et al., 1996*).

## Cell lysis, Western blotting, and sucrose gradient sedimentation

To immunoblot for Num1-13myc, Dyn1-TAP, Jnm1-13myc, Nip100-13myc, and CAP-GlyΔ−13myc, yeast strains were grown overnight in 5 ml of rich media (YPD) at 30°C. Cell pellets were resuspended in ice cold lysis buffer containing 20 mM Tris pH 7.5, 150 mM NaCl, 1 mM EDTA, 1.5% Triton X-100, supplemented with protease inhibitor cocktail tablet (Millipore Sigma). Equal amount of cells were lysed by bead beating in round-bottom glass tubes for 6 × 30 s with 2 min interval between beatings. Following centrifugation (at 21,130 g for 10 min at 4°C), the resulting supernatants were separated on 8% (for Num1-13myc and Jnm1-13myc) or 6% (for Dyn1-TAP) or 4–15% (for Nip100-13myc and CAP-GlyΔ−13myc) SDS-PAGE and then electro-blotted to PVDF or nitrocellulose membrane in 25 mM Tris, 192 mM glycine, 0.05% SDS, and 20% methanol for 80 min. Membranes were probed with either mouse 9E10 anti-c-Myc antibody (BioLegend) at 1:250 or 1:500 or 1:1000 dilution, or rabbit IgG antibody (GenScript) at 1:5000 dilution. Goat HRP-conjugated anti-mouse (BioLegend) and anti-rabbit antibodies (Jackson ImmunoResearch) were used at 1:10,000 dilutions. Chemiluminescence signals were acquired and imaged using a ChemiDoc Imaging System (Bio-Rad) or a G:BOX Chemi HR16 (Syngene) equipped with a 16-bit CCD camera (Sony ICX285AL; pixel size of 6.45 x 6.45 µm). Immunoblots were exposed for durations ranging from 3 s to 5 min without saturating the camera's pixels.

For sedimentation analysis of Num1-13myc, we poured 10 ml 20–60% sucrose step gradients and allowed them to equilibrate for 9 hr at 4°C before use. Each step of the gradient contained 2 ml of 20, 30, 40, 50, or 60% sucrose in sedimentation buffer (10 mM Tris pH 7.5, 10 mM EDTA, and 50 mM NaCl). WT and *scs2/22Δ* strains expressing Num1-13myc were grown to mid-log phase in 60 ml of YPD media, collected by centrifugation, and resuspended in ice cold lysis buffer containing 20 mM Tris pH 7.5, 1 mM EDTA, and 50 mM NaCl supplemented with protease inhibitor cocktail tablet. Cells were then lysed by glass bead beating for 6 times 30 s with 2 min intervals between beatings. Lysates were clarified at 500 g for 10 min at 4°C and 0.5 ml of the supernatant was loaded directly onto a 10 ml sucrose gradient prepared as above. Centrifugation was performed in a Beckman SW41 Ti rotor at 36,000 rpm for 17.5 hr at 4°C. Fractions of 0.5 ml were collected from the top of each gradient for analysis by Western blot using the mouse 9E10 anti-c-Myc antibody (for Num1-13myc) and the anti-Sac1 antibody (a kind gift from Dr. Charles Barlowe for detection against the ER marker Sac1).

## Statistical methods

All statistical analyses were performed using GraphPad Prism software. A two-tailed Student's *t* test or one-way ANOVA test was used to determine statistical significance where indicated. At least two independent experiments were performed for each analysis.

## Acknowledgements

We thank Rachael Judson, John Beckford, and Ao Liu for valuable help with making yeast strains. We are very grateful to Drs. Thomas Maresca, Patricia Wadsworth, and John Lopes (SO dissertation committee members) for stimulating discussions of our data. This work was supported by NIH/ NIGMS grant (GM076094) to WL Lee and in part by a fellowship from the University of Massachusetts to SO as part of the Biotechnology Training Program funded by National Research Service Award T32 GM108556. SG is supported in part by a Dartmouth College Undergraduate Presidential Scholar Assistantship.

## Additional information

### Funding

| Funder | Grant reference number | Author |
| --- | --- | --- |
| University of Massachusetts Amherst | Graduate Student Fellowship as part of the T32 Biotechnology Training Program GM108556 | Safia Omer |
| Dartmouth College | Undergraduate Presidential Scholar Assistantship | Samuel R Greenberg |
| National Institute of General Medical Sciences | GM076094 | Wei-Lih Lee |

The funders had no role in study design, data collection and interpretation, or the decision to submit the work for publication.

### Author contributions

Safia Omer, Conceptualization, Data curation, Formal analysis, Validation, Investigation, Methodology, Writing—original draft, Writing—review and editing; Samuel R Greenberg, Investigation, Substantial contributions to acquisition of data as well as analysis and interpretation of data; Wei-Lih Lee, Conceptualization, Supervision, Funding acquisition, Investigation, Writing—original draft, Writing—review and editing

### Author ORCIDs

Safia Omer ![ORCID] http://orcid.org/0000-0002-3009-4496
Samuel R Greenberg ![ORCID] http://orcid.org/0000-0002-4176-3958
Wei-Lih Lee ![ORCID] http://orcid.org/0000-0002-5606-4754

### Decision letter and Author response

Decision letter https://doi.org/10.7554/eLife.36745.038
Author response https://doi.org/10.7554/eLife.36745.039

## Additional files

### Supplementary files

• Supplementary file 1. Yeast strains used in this study.
DOI: https://doi.org/10.7554/eLife.36745.035
• Transparent reporting form
DOI: https://doi.org/10.7554/eLife.36745.036

### Data availability

All data generated or analyzed during this study are included in the manuscript and supporting files. All yeast strains generated in this work are provided in Supplementary File 1.

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
