## [Decision Letter]

Thank you for submitting your article "Cortical dynein pulling mechanism is regulated by differentially targeted attachment molecule Num1" for consideration by *eLife*. Your article has been reviewed by three peer reviewers, and the evaluation has been overseen by Andrea Musacchio as the Senior and Reviewing Editor. The following individuals involved in review of your submission have agreed to reveal their identity: Dimitris Liakopoulos (Reviewer #2); Kerry Bloom (Reviewer #3).

The reviewers have discussed the reviews with one another and the Reviewing Editor has drafted this decision to help you prepare a revised submission.

Summary:

In this manuscript, Omer and Lee find that a *S. cerevisiae* null mutant for the ER tethers Scs2 and Scs22 dramatically alters the cortical distribution of the dynein receptor Num1 without causing misalignment of anaphase spindles. Cells without Scs2/22 switch their spindle positioning mechanism from predominantly microtubule sliding to predominantly capture-shrinkage, which is mediated by a Num1 patch at the bud tip where dynein is off-loaded. Further experiments show that the capture-shrinkage pathway is not dependent on mitochondria in the bud (which were previously implicated in Num1 patch formation at the cortex), nor on the kinesins Kip3/Kar3, but that it requires dynein motor activity and microtubule binding by the CAP-Gly domain of the dynactin subunit Nip100. Importantly, the authors are able to re-instate the sliding mechanism in Scs2/22 null cells by creating lateral Num1 patches on the cortex via a Num1-CAAX motif fusion. The authors propose an attractive model in which dynein-dependent microtubule sliding and capture-shrinkage are used cooperatively for efficient spindle positioning. The study further illustrates that the spatial distribution of cortical dynein receptors such as Num1 determines the relative contribution of lateral versus end-on cortical microtubule contacts to spindle positioning. This is an elegant and convincing study, executed to a high technical standard.

Essential revisions:

a) The existence of different Num1 patches is a major conclusion of the paper, but what makes them different remains unclear. The authors could at least start to address this question by exploring one or more of the following proposed avenues:

1) Measure localization intensity of other components of the dynein pathway like Jnm1 on the Num1 patches as well as co-localization of the patches in *scs* with other polarized factors (exosome and polarisome) at the bud tip. It could be that the end-on pulling patches are specifically the ones at the tip, at polarisome or exosome sites.

2) Test whether the Num1 patches in *scs* mutants can support spindle elongation by examining synthetic lethality with *cin8∆* mutants (i.e. are *scs2/22∆ cin8∆* or *kar9∆ scs2/22∆ cin8∆* mutants growth defective?). This may help differentiate between the functions of the different types of patches.

3) Examine whether all Num1-CAAX patches contain dynein, Jnm1 (or the dynein C-terminal fragment that localizes to Num1 dots) – because in *scs* Num1-CAAX cells there is still a fraction of cells in which MTs interact end-on (Figure 5C).

4) Overexpress Num1 in *scs* mutants and test, whether the Num1 patch intensity of the dimmer patches as well as the MT sliding phenotype can be restored. This would show that one difference between patches is their Num1 content.

5) Correlate between the position of the patch at the cortex and the MT behavior (sliding vs. end on pulling), in order to address the dependence of the mechanism of pulling on the geometry of the MT-Num1 patch interaction.

b) It should be made clearer which experiments are made in a *kar9∆* background; the precise phenotype should be declared on the figures/graphs/images. For example, it is not clear whether the experiments using the dyneinK24224A mutant or the Num1-GFP-CAAX variant are made in a *kar9∆* background (that shouldn't be possible for dyneinK24224A) and thus how to compare this results with the other strains.

c) In the graph of Figure 1B, the fraction of cells containing 1, 3, 5, 7,… bright cortical foci is plotted. Plotting the full distribution (1, 2, 3, 4, 5,.…foci) would correlate better with the description of the results: 'More than 70.0% of *scs2/22Δ* budded cells displayed ≤ 2 bright patches…'

d) In Figure 1—figure supplement 5, the authors show that preventing mitochondrial segregation into the bud does not affect the Num1 patch at the bud tip in Scs2/22 null cells. The authors' model is that this Num1 patch is primarily responsible for the capture-shrinkage mechanism that is sufficient for correct spindle positioning. By contrast, recent work from Kraft and Lackner (2017) showed that spindles are mispositioned in mutants that prevent mitochondrial inheritance. Can the authors comment on this apparent discrepancy?

e) Following the analysis of the *num1^LL^* mutant, which prevents off-loading of dynein (and presumably dynactin) to the cortex, the authors state in the Results (subsection “Dynein mediates spindle movements via capture-shrinkage of astral MTs at the bud tip): 'These results contradict the model in which dynein does not need to attach to the cortex to destabilize MT ends (Estrem et al., 2017).'

The authors should clarify why they think their results contradict those of Estrem et al. (2017). The model proposed by Estrem et al. postulates that after cortical off-loading of dynein-dynactin, some dynein is left at the microtubule tip without dynactin (all of which is off-loaded), which then results in destabilization of the microtubule. If off-loading is prevented, such as in the *num1^LL^* mutant, dynactin would always be present at the tip together with dynein and thus microtubules would not be destabilized. Therefore, I don't really see the contradiction.

f) In Figure 4, the CAP-Gly domain of the dynactin subunit Nip100 is deleted, which results in less capture-shrinkage events. The authors should determine the expression levels of the deletion mutant by western blot to exclude the possibility that the defects are due to an overall reduction in Nip100 levels.

---

## [Author Response]

Essential revisions:a) The existence of different Num1 patches is a major conclusion of the paper, but what makes them different remains unclear. The authors could at least start to address this question by exploring one or more of the following proposed avenues:1) Measure localization intensity of other components of the dynein pathway like Jnm1 on the Num1 patches as well as co-localization of the patches in scs with other polarized factors (exosome and polarisome) at the bud tip. It could be that the end-on pulling patches are specifically the ones at the tip, at polarisome or exosome sites.

We have now addressed 4 out of 5 of the proposed avenues. First, as suggested, we have measured the localization intensity of Jnm1, which is presented in the new Figure 2D and Figure 2—figure supplement 1B. The results show that, like Dyn1-3GFP, Jnm1-3mCherry was enhanced at the MT plus end and cortex in *scs2/22∆* relative to WT cells. Additionally, as suggested, we have included new experiments to address the localization of Num1 at the bud tip in *scs2/22∆* cells. We discovered that deletion of *BNI1* in *scs2/22∆* cells caused a significant decrease in observing Num1-GFP patches at the bud tip (new Figure 6A and B) resulting in a dramatic spindle misalignment phenotype (new Figure 6C). We also show that disruption of F-actin using latrunculin A did not perturb Num1-GFP patches at the bud tip (new Figure 6D), indicating that the actin nucleation activity of Bni1 is not required for the maintenance of Num1 patches at this site. Furthermore, we show that deletion of *BNI1* in *kar9∆* cells resulted in a significant increase in astral microtubules that slid past the bud tip during spindle correction (new Figure 7B and C). This phenotype lends further support to our model that the population of Num1 that localizes to the bud tip functions as a brake for MT sliding. Importantly, the population of Num1 at the bud tip is different from the Scs2/22-dependent population, whose localization along the cell periphery did not appear to be affected by *BNI1* deletion (new Figure 7A), further strengthening the main conclusion of our paper. We thank the reviewer for the push to get these data. Samuel Greenberg helped us collect and analyze the *bni1∆ kar9∆* data presented in Figure 7B and C, and consequently he has been added as an author to the revised manuscript.

2) Test whether the Num1 patches in scs mutants can support spindle elongation by examining synthetic lethality with cin8∆ mutants (i.e. are scs2/22∆ cin8∆ or kar9∆ scs2/22∆ cin8∆ mutants growth defective?). This may help differentiate between the functions of the different types of patches.

We have explored this avenue by performing tetrad dissection analysis. The results show that *cin8∆ scs2/22∆* triple mutant progeny formed viable colonies that were comparable to *cin8∆* single mutant or *scs2/22∆* double mutant (new Table I). As positive control, *cin8∆ dyn1∆* double mutant progeny were inviable. These data did not help differentiate between the functions of the different types of patches, but they show that Num1 patches in *scs2/22∆* cells can support spindle elongation.

3) Examine whether all Num1-CAAX patches contain dynein, Jnm1 (or the dynein C-terminal fragment that localizes to Num1 dots) – because in scs Num1-CAAX cells there is still a fraction of cells in which MTs interact end-on (Figure 5C).

As suggested, we have now examined dynein and dynactin localization in *scs2/22∆* cells expressing Num1-GFP-CAAX (see new Figure 5—figure supplement 1). The results show that cortical Dyn1-3mCherry and Jnm1-3mCherry foci were found at the polar ends of the cell as well as along the cell perimeter in *NUM1-GFP-CAAX scs2/22∆* cells, compared with only being found at the polar ends of the cell in *NUM1-GFP scs2/22∆* background. However, we did not anticipate finding all Num1-GFP-CAAX patches to contain dynein and dynactin, since we have previously demonstrated that not every single Num1-GFP patches in WT cells contains an offloaded cortical dynein and dynactin patch (see Figure 3A in Markus et al., Curr Biol 2009 and Figure S5 in Markus and Lee, Dev Cell 2011). For this reason, we think that the new results did not help differentiate between the different types of Num1 patches. Nonetheless, they show that cortical dynein and dynactin are restored by Num1-GFP-CAAX along the cortex in *scs2/22∆* cells to an extent similar to that seen in WT cells expressing Num1-GFP, consistent with Num1-GFP-CAAX re-instating the sliding mechanism in *scs2/22∆* cells.

4) Overexpress Num1 in scs mutants and test, whether the Num1 patch intensity of the dimmer patches as well as the MT sliding phenotype can be restored. This would show that one difference between patches is their Num1 content.

We did not pursue this avenue because, in our hands, overexpression (e.g. using a *GAL1* promoter) can often result in a very high variability in the expression levels, making it difficult to compare phenotype from cell to cell. Another pitfall of overexpression is that the localization of the overexpressed protein might not reflect the physiological localization of the endogenous protein. Because of these concerns, we believe that it would be difficult to interpret the change in Num1 intensity and function.

5) Correlate between the position of the patch at the cortex and the MT behavior (sliding vs. end on pulling), in order to address the dependence of the mechanism of pulling on the geometry of the MT-Num1 patch interaction.

As suggested, we now have correlated the position of the MT-Num1 interaction with the MT behavior observed in the bud. The results show that sliding is correlated with MT interacting with a Num1 patch located within the proximal three quarters of the bud cortex, whereas end-on pulling is correlated with MT interacting with a Num1 patch located within the distal quarter of the bud. These results are presented in the new Figure 5D.

b) It should be made clearer which experiments are made in a kar9∆ background; the precise phenotype should be declared on the figures/graphs/images. For example, it is not clear whether the experiments using the dyneinK24224A mutant or the Num1-GFP-CAAX variant are made in a kar9∆ background (that shouldn't be possible for dyneinK24224A) and thus how to compare this results with the other strains.

We apologize for this oversight. In the revised figures, we have now indicated the full genotype on all relevant graphs and images, making it clear which experiments were performed in the *kar9∆* background. We have also made the corresponding changes in the revised text. Thanks for pointing this out to us.

c) In the graph of Figure 1B, the fraction of cells containing 1, 3, 5, 7,… bright cortical foci is plotted. Plotting the full distribution (1, 2, 3, 4, 5,.…foci) would correlate better with the description of the results: 'More than 70.0% of scs2/22Δ budded cells displayed ≤ 2 bright patches…'

As requested, we have re-plotted the graph to show the full distribution of foci in Figure 1B. Additionally, we have similarly re-plotted the distribution in Figure 1—figure supplement 1D. Thank you for pointing this out to us.

d) In Figure 1—figure supplement 5, the authors show that preventing mitochondrial segregation into the bud does not affect the Num1 patch at the bud tip in Scs2/22 null cells. The authors' model is that this Num1 patch is primarily responsible for the capture-shrinkage mechanism that is sufficient for correct spindle positioning. By contrast, recent work from Kraft and Lackner (2017) showed that spindles are mispositioned in mutants that prevent mitochondrial inheritance. Can the authors comment on this apparent discrepancy?

Thanks for the opportunity to clarify this. We think that the discrepancy might be due to a differential effect of cortical ER and mitochondria on the different populations of Num1 (Scs2/22-dependent, mitochondria-dependent, and Bni1/bud tip-dependent). To clarify, our analysis shows that the Num1 patch at the bud tip was not affected by mitochondrial inheritance (Figure 1—figure supplement 5) *and* was significantly enhanced in the *scs2/22* null background (due to redistribution of Num1 to the bud tip when cortical ER tethering is disrupted) (Figure 1—figure supplement 1F). We also showed that it contains an elevated level of offloaded dynein and dynactin in the *scs2/22* null background (Figure 2C and new Figure 2D). Thus, we speculate that the patch at the bud tip might exert a larger force in *scs2/22* null cells (because the patch contains an enhanced level of Num1 to anchor a higher amount of dynein/dynactin), explaining why it is sufficient for correct spindle positioning. By contrast, the ‘mitochondria-assembled Num1 clusters’ described by Kraft and Lackner (2017) most likely correspond to the bright Num1 patches that we observed along the cell periphery (compare Figure 1A in Kraft and Lackner with Figure 1A and Video 1 in our paper), not the Num1 patch at the bud tip. Their data did not show redistribution of Num1 to the bud tip when mitochondrial inheritance was prevented. Because the patch at the bud tip was not enhanced (and consequently did not contain elevated levels of offloaded dynein and dynactin), it failed to correct spindle positioning in their mutants that were defective in mitochondrial inheritance. Thus, we interpreted that cortical ER and mitochondria may have different effects on Num1 distribution when these organelles are prevented from interacting with the cortex.

e) Following the analysis of the num1^LL^ mutant, which prevents off-loading of dynein (and presumably dynactin) to the cortex, the authors state in the Results (subsection “Dynein mediates spindle movements via capture-shrinkage of astral MTs at the bud tip): 'These results contradict the model in which dynein does not need to attach to the cortex to destabilize MT ends (Estrem et al., 2017).'The authors should clarify why they think their results contradict those of Estrem et al. (2017). The model proposed by Estrem et al. postulates that after cortical off-loading of dynein-dynactin, some dynein is left at the microtubule tip without dynactin (all of which is off-loaded), which then results in destabilization of the microtubule. If off-loading is prevented, such as in the num1^LL^ mutant, dynactin would always be present at the tip together with dynein and thus microtubules would not be destabilized. Therefore, I don't really see the contradiction.

To clarify, our statement referenced above refers to whether dynein needs to attach to the cortex to mediate MT depolymerization. We would like to point out that Estrem et al. (2017) stated that dynein *does not* need to attach to the cortex to depolymerize MT ends (see Estrem et al. (2017) Discussion, subsection “Dynein motor activity destabilizes microtubules”). They found that the duration of plus end interactions at the cortex was shorter-lived in the *num1∆* mutant compared with that in the *dyn1∆* mutant (see Figure 3C in Estrem et al., 2017), which they interpreted as dynein being able to destabilize MT ends in the *num1∆* background, where offloading was prevented. In our opinion, this evidence is in conflict with their proposed model, since, as reasoned by the reviewer above, dynactin would be present at the MT tip together with dynein (because there is no offloading in *num1∆*), and thus the MT would not be destabilized. In our study, we used the *num1^LL^* allele to prevent offloading and found that cortical attachment is indeed required for dynein to destabilize MT (Video 4). Thus, we stated that our results contradict the conclusion made by Estrem et al. (2017). Nonetheless, in order to make it clearer, we have now modified our statement to read: “These results contradict a previous study postulating that dynein does not need to attach to the cortex to destabilize MT ends”.

f) In Figure 4, the CAP-Gly domain of the dynactin subunit Nip100 is deleted, which results in less capture-shrinkage events. The authors should determine the expression levels of the deletion mutant by western blot to exclude the possibility that the defects are due to an overall reduction in Nip100 levels.

As requested, we have tagged the CAP-Gly∆ construct with 13myc and assessed its expression levels by western blot. The results are presented in the new Figure 4B and described in the revised text (subsection “CAP-Gly domain of Nip100/p150^Glued^ is required for dynein-mediated capture-shrinkage of astral MTs”, first paragraph). The level of the truncated Nip100 protein in *scs2/22∆* cells was similar to that of the full-length Nip100 protein in WT cells, indicating that the observed reduction in capture-shrinkage events could not be due to an overall reduction in protein stability. Thanks for the suggestion to get these data.